# The transcription factor ChREBP Orchestrates liver carcinogenesis by coordinating the PI3K/AKT signaling and cancer metabolism

Emmanuel Benichou[1,8], Bolaji Seffou[1,8], Selin Topçu[1,8], Ophélie Renoult [2], Véronique Lenoir[1], Julien Planchais [1], Caroline Bonner [3,4,5,6], Catherine Postic [1], Carina Prip-Buus[1], Claire Pecqueur [2], Sandra Guilmeau [1], Marie-Clotilde Alves-Guerra [1] & Renaud Dentin [1,7] ✉

Cancer cells integrate multiple biosynthetic demands to drive unrestricted proliferation. How these cellular processes crosstalk to fuel cancer cell growth is still not fully understood. Here, we uncover the mechanisms by which the transcription factor Carbohydrate responsive element binding protein (ChREBP) functions as an oncogene during hepatocellular carcinoma (HCC) development. Mechanistically, ChREBP triggers the expression of the PI3K regulatory subunit p85α, to sustain the activity of the pro-oncogenic PI3K/AKT signaling pathway in HCC. In parallel, increased ChREBP activity reroutes glucose and glutamine metabolic fluxes into fatty acid and nucleic acid synthesis to support PI3K/AKT-mediated HCC growth. Thus, HCC cells have a ChREBP-driven circuitry that ensures balanced coordination between PI3K/AKT signaling and appropriate cell anabolism to support HCC development. Finally, pharmacological inhibition of ChREBP by SBI-993 significantly suppresses in vivo HCC tumor growth. Overall, we show that targeting ChREBP with specific inhibitors provides an attractive therapeutic window for HCC treatment.

Primary liver cancer is a global public health concern as it represents one of the most common and lethal *human* malignancies observed worldwide. Hepatocellular carcinoma (HCC) accounts for more than 80% of primary liver cancers and represents the third most common cause of cancer mortality. While surgical resection and liver transplantation are effective options in the treatment of early-stage disease, therapeutic approaches for advanced HCC are very limited. As a matter of fact, the history of systemic treatment for HCC looks like a battlefield with few survivors as a result of a positive outcome[1]. In the past, only multikinase inhibitors, such as sorafenib in first line and regorafenib in second line were approved for late-stage HCC treatment, with unfortunately modest benefits for the patients in terms of survival[1]. However, immunotherapies such as immune checkpoint inhibitors (ICIs) have revolutionized the management of HCC in the past 5 years. For example, the combination of atezolizumab (an anti-PDL1 antibody) and bevacizumab (anti-VEGFA antibody) has produced superior results when compared with sorafenib in patients with advanced stage HCC, setting a new first line benchmark median overall survival duration of

[1]Université Paris Cité, Institut Cochin, INSERM, CNRS, F-75014 Paris, France. [2]Nantes Université, INSERM U1307, CNRS 6075, CRCI2NA Nantes, France. [3]Institut Pasteur de Lille, Lille, France. [4]INSERM, U1011 Lille, France. [5]European Genomic Institute for Diabetes, Lille, France. [6]Université de Lille, Lille, France. [7]Present address: Institut Cochin, Faculté de Médecine 3ème étage, 24 Rue du Faubourg Saint Jacques, 75014 Paris, France. [8]These authors contributed equally: Emmanuel Benichou, Bolaji Seffou, Selin Topçu. ✉e-mail: renaud.dentin@inserm.fr

19 months, and thus constitutes a breakthrough in the management of this disease[2]. Unfortunately, immunotherapies do not provide a definitive cure for patients with advanced stage HCC and around 60% of patients still do not respond to this treatment. Thus, the elucidation of the molecular pathogenesis of HCC is still crucial for developing more effective alternative therapeutic strategies.

Because specific metabolic features are altered across many cancer types, reprogrammed metabolism is considered a hallmark of cancer[3]. Specifically, changes in tumor bioenergetics, consisting of elevation of glycolysis, upregulation of lipid and amino acid metabolism and induction of the pentose phosphate pathway (ppp), are detected in most cancer types and often negatively correlated with survival prognosis[4]. Tumor cells use glycolysis to meet their metabolic needs regardless of oxygen concentrations, a phenomenon known as the "Warburg effect" or aerobic glycolysis[5]. In HCC, the dysregulation of two specific glycolytic enzymes known as hexokinase 2 (HK2) and pyruvate kinase M2 (PKM2) allows this metabolic switch required for hepatocyte proliferation and partly for tumorigenesis[6]. As a consequence, excessive aerobic glycolysis is commonly associated with HCC invasiveness and poor prognosis[7]. Furthermore, while energy-providing lipids come almost exclusively from dietary lipids in non-proliferative cells, a marked induction of de novo lipid synthesis (lipogenesis) occurs in cancer cells and often correlates with HCC initiation and cancer progression[8]. Thus, the major differences in bioenergetic features between normal and cancer cells may open up new therapeutic avenues for targeting these related metabolic pathways selectively in the early stages of HCC treatment[9]. Overall, the key question driving research in the field is to identify key metabolic candidates whose inactivation will impair hepatocarcinogenesis while sparing normal cells for therapeutic benefits.

In this context, we have previously established that the glucose responsive transcription factor Carbohydrate Responsive Element Binding Protein (ChREBP) plays a central role in the regulation of multiple metabolic pathways in nonproliferating hepatocytes[10,11]. ChREBP is a major mediator of glucose action on glycolytic (pklr), PPP (g6pdh, tkt) and lipogenic (acc, fasn) gene expression[12–14]. Therefore, given its importance in regulating hepatic energy metabolism, ChREBP may thus represent a promising candidate for targeted therapies during HCC treatment. This study demonstrates that enhanced hepatic ChREBP activity in mice is sufficient to initiate the development and progression of HCC, unraveling its oncogenic function in the liver. At the molecular level, our study unravels that ChREBP enhances the PI3K/AKT (phosphatidylinositol-3-kinase/Ak strain transforming) signaling in a p85α-dependent manner, which in turn contributes to its pro-proliferative effect. In addition, ChREBP coordinately rewires both glucose and glutamine metabolic fluxes to enhance de novo nucleotide and fatty acid synthesis to sustain cell proliferation. Altogether, these results support a novel mechanism by which ChREBP activation contributes to tumorigenicity during HCC development. Finally, the characterization of the small molecule SBI-993 as a potent inhibitor of ChREBP activity, cell proliferation and adjuvant of sorafenib treatment, demonstrates that ChREBP represents a strong candidate for pharmacological intervention during HCC treatment.

## Results

### ChREBP expression is increased in *human* HCC and is associated with tumor aggressiveness

The clinical relevance of ChREBP expression levels during *human* liver carcinogenesis was first evaluated by assessing the relationship between its expression and patient's clinicopathological records in two independent HCC cohorts, namely LIHC and LICA-FR, for which clinical data were available (Supplementary Tables 1 and 2 and Supplementary Data 1 and 2). When compared to corresponding normal tissues, both the LIHC and LICA-FR datasets show high levels of ChREBP gene expression within the tumor (Fig. 1a). In addition, the expression of specific and well-described ChREBP-regulated genes is also increased within the tumor, supporting enhanced ChREBP activity during HCC development (Fig. 1b). Of note, the correlation between ChREBP expression with a list of 40 genes, previously validated for their association with good or poor HCC prognosis[15], demonstrated that high ChREBP mRNA levels systematically sign HCC tumors with poor prognosis (Fig. 1c). Supporting this result, ChREBP expression was significantly more elevated in HCC samples with poor prognosis when compared with those having a better outcome based on clinical data available from these datasets (Fig. 1d). Accordingly, increased ChREBP expression within the tumor correlates with lower patient's survival rate in both LIHC and LICA-FR cohorts (Fig. 1e). Transcriptome profiling further revealed that ChREBP expression was specifically increased in HCC from patients with chronic liver diseases including viral HCV infection, NAFLD and NASH etiologies (Supplementary Fig. 1a, b). This demonstrates that HCC tumors, progressing on HCV, NAFLD and NASH affected livers and for which ChREBP expression is elevated, accumulated on the higher risk side. In opposite, HCC tumors which progressed on HBV infection or chronic alcohol consumption (ALD), and for which ChREBP expression is respectively decreased or not altered compared to non-affected HCC, accumulated on the lower risk side (Supplementary Fig. 1a, b). Furthermore, the analysis of 8 additional publicly available datasets (Supplementary Table 1)[16–23] confirmed that elevated ChREBP mRNA and its target genes were significantly increased in all HCC datasets (Supplementary Fig. 2). In addition, Western blot analysis of 12 pairs of *human* HCC (T) and corresponding adjacent normal liver tissue (N) revealed that ChREBP protein levels were also increased in most of the tumor samples (Fig. 1f and supplementary Table 3). Finally, the expression of MondoA, a paralog of ChREBP that also senses glycolytic flux in skeletal muscle[24], was analyzed in all these 10 *human* HCC datasets[25]. In contrast to ChREBP, MondoA mRNA and protein levels were not altered within the tumor compared with corresponding normal tissues (Fig. 1f and Supplementary Figs. 1c and 2). Overall, these findings revealed, among the Mondo glucose-responsive transcription factors, a specific link between ChREBP expression levels and poor prognosis HCC development in *humans*.

### Stable hepatic ChREBP overexpression promotes HCC initiation and development

To test its potential oncogenic function, ChREBP was specifically overexpressed in the liver of male C57BL6/J *mice*. Three weeks after adenoviral ChREBP delivery, *mice* displayed hepatic steatosis, that was characterized by increased liver weight and size (Fig. 2a). Interestingly, this liver hypertrophy was associated with an increased hepatocyte proliferation rate, as evidenced by higher number of BrdU positive hepatocytes and enhanced expression of specific cell cycle proteins (Fig. 2b, c). While this suggested that short-term increased in ChREBP activity was sufficient to trigger hepatocyte hyperproliferation, ChREBP contribution to the initiation and/or progression of hepatocarcinogenesis was further investigated. Therefore, ChREBP (HA-tagged) was stably overexpressed in the liver of FVB/N *mice* by combining hydrodynamic injection with the use of the non-viral sleeping beauty (SB) transposon system (Supplementary Fig. 3a). The firefly luciferase (empty vector) was stably overexpressed alone (Ctrl) or with ChREBP to follow hepatocyte proliferation and tumor development. Successful integration of the luciferase in both cases was validated 7 days post-injection (Supplementary Fig. 3b). While overexpression of the empty vector did not induce changes in luciferase activity over 42 weeks (Fig. 2d), stable ChREBP overexpression led to a time-dependent increase in bioluminescence activity, suggesting high proliferation rate in ChREBP-transduced hepatocytes (Fig. 2d). Over time, ChREBP promoted the initiation and development of small hepatocellular adenomas, that gradually progress to HCC between 6 and 12 months in 100% of treated *mice* (Fig. 2e). ChREBP overexpressing tumors displayed

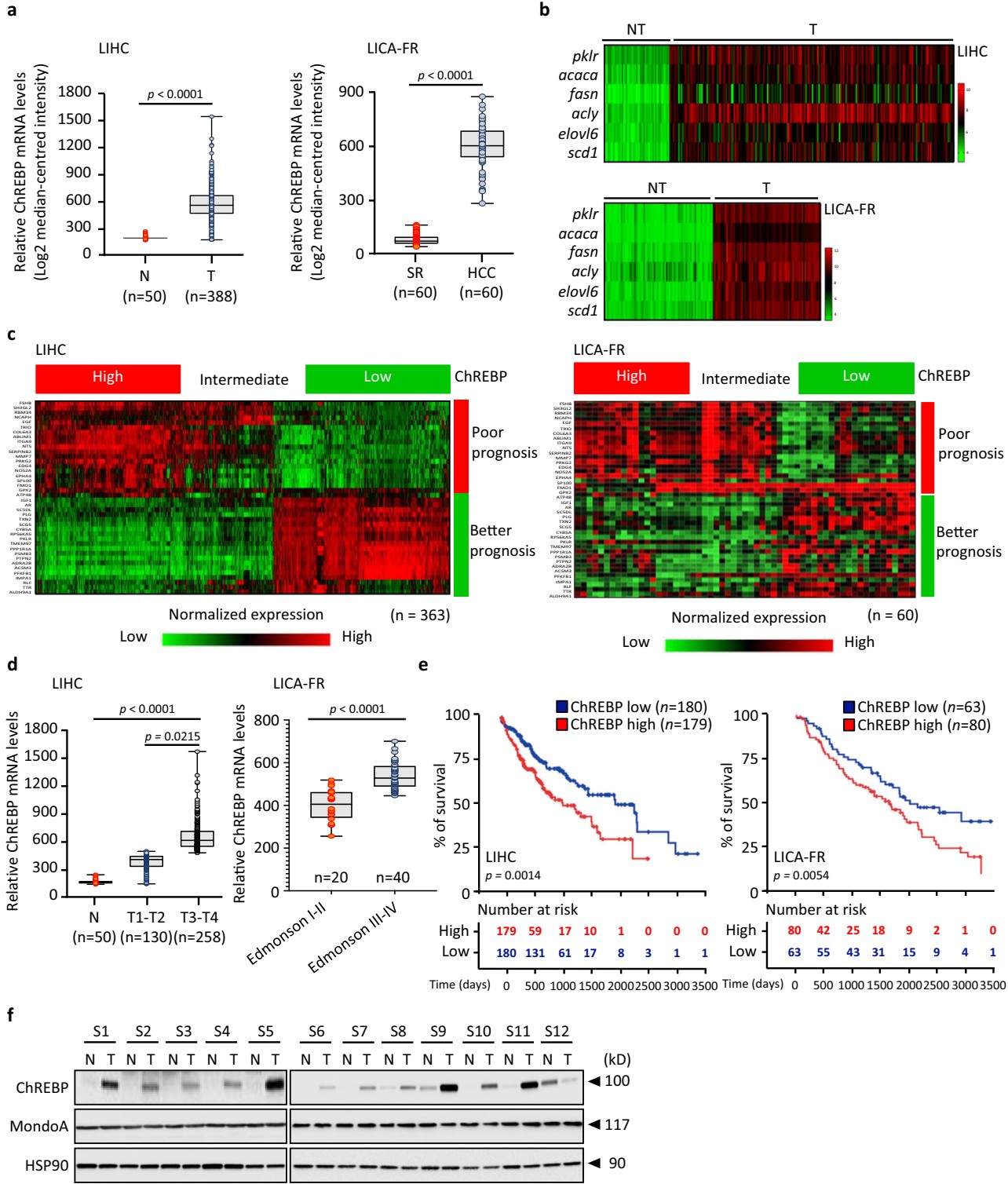

high lipid content, in agreement with the role of ChREBP in regulating de novo lipid synthesis (Fig. 2f), as well as sustained proliferation index, as shown by the increased number of Ki67 positive hepatocytes (Fig. 2g and Supplementary Fig. 3c) and the enhanced expression of cell cycle regulators (Fig. 2h). Importantly, as previously described in *human*'s HCC, no change in MondoA mRNA and protein levels was observed in response to ChREBP hyperactivation in these two models (Supplementary Fig. 3d, e). All together, these results provided in vivo evidence that ChREBP constitutes a key driver in HCC initiation and development.

## ChREBP tumors cluster *human* HCC subclasses with poor prognosis signatures

To identify the early events by which enhanced ChREBP transcriptional activity favor HCC development, bioinformatic analysis of transcriptomic datasets from liver overexpressing ChREBP for 3 weeks (pre-malignant model) and from ChREBP overexpressing tumors (malignant model) was performed. This analysis first led to the identification of differentially expressed genes respectively in the pre-malignant (GFP versus ChREBP) and tumoral (adjacent non-tumoral versus tumoral tissues) models. Then, common differentially

**Fig. 1 | ChREBP expression is increased in *human* HCC and signs tumors with poor prognosis. a** Data mining of ChREBP gene expression level between HCC (T) and normal liver tissues (N) from the LIHC and LICA-FR datasets relative to the TBP gene expression (Supplementary Data 1 and 2 and Source Data). **b** Heatmap showing the expression of well-described ChREBP-regulated genes in the LIHC and LICA-FR datasets relative to the TBP gene expression. **c** Heatmap of a 40-HCC gene signature, which classified the patients from the LIHC and LICA-FR datasets with either poor or better prognosis depending on ChREBP expression within the tumors relative to the TBP gene expression. ChREBP expression was divided into tertiles based on low, intermediate, or high expression levels. **d** ChREBP expression in HCC based on individual tumor grade from the LIHC and LICA-FR datasets relative to the TBP gene expression. **e** Kaplan–Meier analysis from the LIHC and LICA-FR Oncomine datasets depicting the overall survival rate of patients with low or high ChREBP expression levels within tumors (mRNA level, bottom 50% vs. top 50%). The tables shown below the Kaplan–Meier survival curves listed the number of patients at risk at a specific time point. **f** Expression profile of ChREBP and MondoA protein contents in 12 paired HCC (T) and adjacent non-tumoral tissues (N) (clinical characteristics of patients can be found as Source data provided as a Source Data file) ($n = 12$). **a, d** For all box plots, the boundary of the box closest to zero indicates the 25th percentile, a black line within the box marks the median, and the boundary of the box farthest from zero indicates the 75th percentile. Whiskers above and below the box indicate the 10th and 90th percentiles. Points above and below the whiskers indicate outliers outside the 10th and 90th percentiles (statistical analysis can be found in the Source Data file). **a, d** Statistical analyses were made using unpaired two-sided Student's $t$ test. **e** Significant difference in survival between cohorts was calculated using the log-rank (Mantel Cox) test. Source data are provided as a Source Data file.

upregulated genes between these two models were selected. This allowed the identification of a specific signature, including 324 genes shared between the two models, that may explain the primary oncogenic role of ChREBP in the liver (Supplementary Fig. 3f and Supplementary Data 3). These genes are specifically involved in cell proliferation, cell cycle, cell division or metabolic processes, all known markers of HCC (Fig. 2i and Supplementary Fig. 3g). Accordingly, GSEA analysis revealed an enrichment in gene signatures specific for HCC subclasses previously associated with a proliferative and invasive phenotype together with a worse outcome (Fig. 2j and Supplementary Data 4). In addition, we found a significant enrichment in tumor signatures associated with invasion, stemness and poor prognostic features (Fig. 2j and Supplementary Data 4). Finally, the expression of genes known to be associated with HCC with poor prognosis in *human*, as well as the expression of fetal genes that signed undifferentiated tumors, were also significantly induced in all ChREBP tumors (Supplementary Fig. 3h). As a result, ChREBP-mediated HCC initiation and progression correlated with lower *mice* survival rate (Fig. 2k). Taken together, our results demonstrate that ChREBP acts as an oncogene to promote HCC initiation and development. Furthermore, the genomic profile of these ChREBP tumors recapitulates those of proliferative subclasses of *human* HCC with a poor clinical outcome.

## The induction of the PI3K regulatory subunit p85α drives the stimulatory effects of ChREBP on PI3K/AKT signaling and hepatocyte proliferation

Additional bioinformatic analysis identified the IR/IGF1R receptor-mediated regulation of the pro-oncogenic PI3K/AKT signaling as the most affected pathway in ChREBP tumors (Supplementary Fig. 4a and Supplementary Data 5 and 6). Western blot analysis and immunostaining of liver sections validated that ChREBP overexpression enhanced the PI3K/AKT signaling in both the pre-malignant and malignant models (Fig. 3a and Supplementary Fig. 4b and c). Therefore, to determine whether enhanced ChREBP activity drives hepatocyte proliferation in a PI3K/AKT-dependent manner, we conducted experiments using the selective allosteric AKT inhibitor MK-2206[26] in ChREBP overexpressing *mice* (pre-malignant model). Upon MK-2206 treatment, ChREBP overexpression was no longer able to stimulate AKT signaling and to enhance the expression of cell cycle genes (Fig. 3b and Supplementary Fig. 4d) and consequently cell proliferation (Fig. 3c, d). In HepG2 and Huh7 hepatoma cell lines stably overexpressing ChREBP, the blockade of ChREBP-induced PI3K/AKT signaling by MK-2206 also significantly impaired the capacity of ChREBP to increase cell proliferation in vitro (Fig. 3e–g and Supplementary Fig. 4e–g). We next addressed whether the expression of direct modulators of the PI3K/AKT signaling, shown to drive tumor growth in cancer, could be directly controlled by ChREBP. To characterize these early ChREBP-regulated genes involved in liver carcinogenesis, ChREBP ChIP-sequencing experiment was performed in primary cultured hepatocytes. This led to the identification of 18,746 ChREBP binding sites, which about half of them (48.95%) were found within the genes (Supplementary Fig. 4h). These binding sites were enriched in the close vicinity region of the transcription start site (TSS) of 3826 potential ChREBP target genes (Supplementary Fig. 4i). Among them, 1367 were also H3K4me3 and RNA-polII positive, suggesting that they were either "poised" for transcription or transcriptionally active (Supplementary Fig. 4j). Furthermore, for genes enriched in the top 1000 intervals, the insulin signaling pathway was among the top 3 pathways with significant number of genes exhibiting ChREBP binding (Supplementary Fig. 4k and l and Supplementary Data 7). Interestingly, among the direct modulators of the PI3K/AKT signaling previously described to play an important role in cancer development, only *Pik3r1*, which encodes the regulatory subunit p85α of the PI3K (Supplementary Fig. 5a), was identified as potential ChREBP regulated gene (Fig. 4a). We confirmed that ChREBP was functionally recruited to the promoter of *Pik3r1* gene in response to high glucose concentrations in primary cultured hepatocytes (Supplementary Fig. 5b). As a result, p85α expression was increased in response to high glucose stimulation and this effect was abolished upon ChREBP silencing (Supplementary Fig. 5c). Further supporting a direct role of ChREBP in controlling *Pik3r1* gene transcription, transient overexpression of ChREBP (WT ChREBP) in HEK293 cells increased the transcriptional activity of a 2.5 kb *Pik3r1* promoter reporter construct whereas this effect was abolished when a dominant negative form of ChREBP (DN ChREBP) was co-overexpressed (Fig. 4b). In agreement, ChREBP recruitment to the promoter of *Pik3r1* was enhanced in ChREBP overexpressing tumors (Fig. 4c), which led to increased p85α expression in these tumors (Fig. 4d). P85α expression was also significantly increased in the pre-malignant model, when ChREBP was overexpressed for 3 weeks in the liver of C57BL/6J male *mice* (Supplementary Fig. 5d).

Under resting conditions, p85α both stabilizes and inactivates the p110 catalytic subunit within the cytosol. However, its inhibitory activity is relieved when p85α is phosphorylated on tyrosine (Tyr458), downstream of multiple growth factor-coupled receptors such as IGF1 (IGF1R) or insulin (IR) receptors (Supplementary Fig. 5a). In this process, the regulatory subunit p85α of the PI3K interacts with the tyrosine-phosphorylated form of GIV (Ga-interacting vesicle-associated protein, also known as Girdin) (Tyr 1764 and Tyr 1798) (Supplementary Fig. 5a). This interaction enables recruitment of tyrosine phosphorylated p85α to the activated forms of IGF1R or IR, stabilized receptor association with PI3K, and enhanced PI3K signaling by increasing the recruitment of AKT at the plasma membrane. Interestingly, the activity of IGF1R and IR was increased in both our pre-malignant and malignant models of ChREBP overexpression as evidence by their increased tyrosine phosphorylation (Supplementary Fig. 5d, e). As a result, tyrosine phosphorylation of p85α and GIV, which signed their respective activation, was also enhanced in response to ChREBP overexpression (Supplementary Fig. 5d, e). Consequently, because p85α directly binds ligand-activated IGF1R

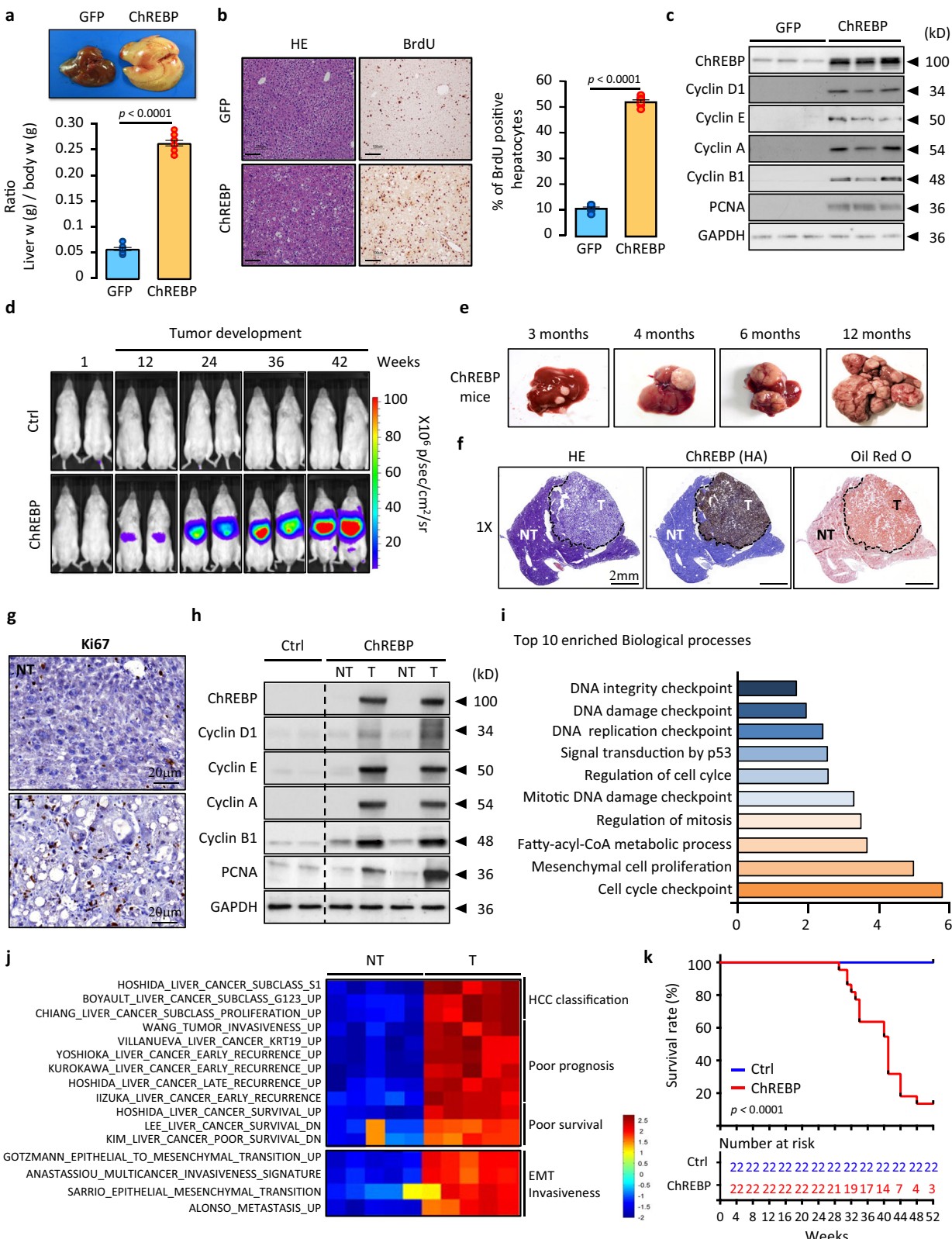

and IR, we next investigated whether ChREBP-mediated increase in p85α expression was able to enhance PI3K activity and facilitate the recruitment of AKT to these activated RTK receptors. In Huh7 cell line, stable ChREBP overexpression potentiated the association of p85α with activated IGF1R and IR upon IGF1 or insulin stimulation (Supplementary Fig. 5f, g). As a result, AKT recruitment to activated IGF1R and IR was enhanced in response to ChREBP overexpression

(Supplementary Fig. 5f, g). Accordingly, PI3K activity, as determined by the ratio between PIP3 and PI(4,5)P2 levels, was increased in ChREBP overexpressing Huh7 cells in response to either IGF1 or insulin stimulation (Supplementary Fig. 5h). More importantly, supporting a stimulatory role of p85α in mediating ChREBP effect on PI3K/AKT signaling, *Pik3r1* gene silencing by CRISPR/Cas9 in ChREBP overexpressing Huh7 cells drastically decreased PI3K activity and

**Fig. 2 | ChREBP overexpression promotes HCC initiation and development in *mice*. a–c** C57BL6/J male *mice* were injected with either GFP or ChREBP overexpressing adenovirus and were studied 3 weeks later. **a** ChREBP overexpressing *mice* exhibit hepatomegaly as shown by the increase in the ratio of liver weigh/body weight (*n* = 6 biologically independent *mice* per group). **b** Representative staining and quantification of liver sections with BrdU from GFP or ChREBP *mice* (*n* = 6 biologically independent *mice* per group). Scale bars = 100 μm. **c** Representative Western blot analysis of proteins of the cell cycle (*n* = 6 biologically independent *mice* per group). **d** Representative bioluminescent imaging depicting tumor development over time after stable ChREBP overexpression (*n* = 10 biologically independent *mice* per group). **e** Representative stepwise development of HCC in ChREBP overexpressing *mice* (*n* = 10 biologically independent *mice* per group). **f** Representative staining of liver sections with H&E, oil red O and specific antibodies against HA-tag (ChREBP) from ChREBP overexpressing *mice* (*n* = 10 biologically independent *mice* per group). NT, non-tumoral; T, tumor. Scale bar = 2 mm. **g** Tumors proliferation index determined by Ki67 immunostaining (representative

shown out of 6 *mice* per group). Scale bar from magnifications of NT and T areas = 20 mm. Quantification is shown in Supplementary Fig. 2c. **h** Western blot analysis of proteins of the cell cycle (*n* = 10 biologically independent *mice* per group). **i** Top ten enriched biological processes from gene ontology analysis performed from a list of 324 genes differentially upregulated in ChREBP tumors (*n* = 12 biologically independent *mice* per group). **j** Heatmap indicating gene-sets significantly affected in ChREBP tumors (*n* = 5 biologically independent *mice* per group). Full list of significantly enriched gene-sets can be found in Source Data File. **k** Kaplan-Meier analysis depicting the survival rate of ChREBP overexpressing *mice* (*n* = 22 biologically independent *mice* per group). The tables shown below the Kaplan–Meier survival curves listed the number of *mice* at risk at the specific time point. Significant difference in survival between groups was calculated using the log-rank (Mantel Cox) test. **a, b** All error bars represent mean ± SEM. Statistical analyses were made using unpaired two-sided Student's *t* test. Source data are provided as a Source Data file.

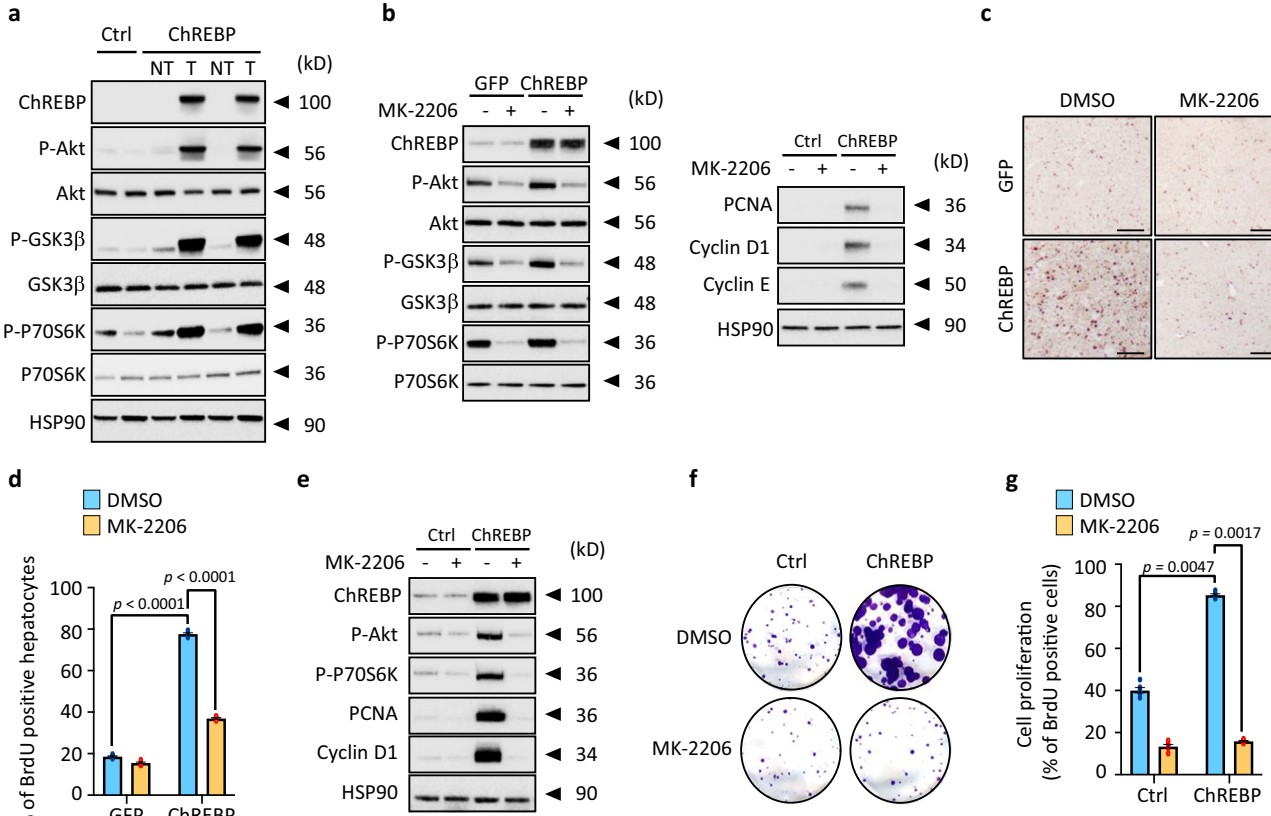

**Fig. 3 | ChREBP overactivation stimulates the pro-oncogenic PI3K/AKT signaling. a** Representative Western blot analysis of the activity of the PI3K/AKT signaling in ChREBP tumors (*n* = 10 biologically independent *mice* per group). **b, c** *Mice*, injected with either GFP or ChREBP overexpressing adenovirus, were orally treated with MK-2206. Captisol (30%) was used as a vehicle for the drug and the control animals were treated with vehicle only. MK-2206 (120 mg/kg) was given orally for 3 weeks on alternate days. **b** Western blot analysis of the PI3K/AKT signaling and proteins of the cell cycle (*n* = 6 biologically independent *mice* per group). **c** Representative staining of liver sections with BrdU. Scale bars = 100 μm

(**d**) Quantification of BrdU staining is shown (*n* = 6 biologically independent *mice* per group). **e–g** HepG2 cells, stably overexpressing ChREBP, were treated with MK-2206 (100 nM) for 24 h. **e** Representative Western blot analysis of proteins of the PI3K/AKT signaling and cell cycle (*n* = 3 independent experiments). **f** Representative clonogenic assay shown (*n* = 3 independent experiments). **g** Cell proliferation index determined by measuring the % of BrdU positive cells (*n* = 3 independent experiments). All error bars represent mean ± SEM. Statistical analyses were determined by two-way analysis of variance (ANOVA) and Tukey's multiple-comparisons test. Source data are provided as a Source Data file.

abolished AKT recruitment to activated IGF1R and IR (Supplementary Figs. 5f–h). Overall, these results demonstrate that ChREBP-mediated regulation of p85α expression is critical for effective formation and stabilization of the PI3K-AKT complex at the plasma membrane. As a result, p85α silencing abolished ChREBP-mediated increase in PI3K/AKT signaling and Huh7 cell proliferation (Fig. 4e, f). Supporting this observation, in our pre-malignant model, inhibiting p85α expression, using adenoviral *pik3r1* shRNA-mediated gene silencing, blunted

ChREBP-mediated hepatocyte proliferation in vivo (Supplementary Fig. 5i, j). Even more strikingly, liver specific CRISPR/Cas9-mediated p85α knockdown in vivo, using hydrodynamic injection, drastically reduced HCC development in *mice* stably overexpressing ChREBP in hepatocytes (Fig. 4g). This protection from HCC development was characterized with reduced PI3K/AKT signaling in tumors (Fig. 4h). Altogether, these results demonstrate that ChREBP-mediated regulation of *Pik3r1* gene transcription drives the development and/or

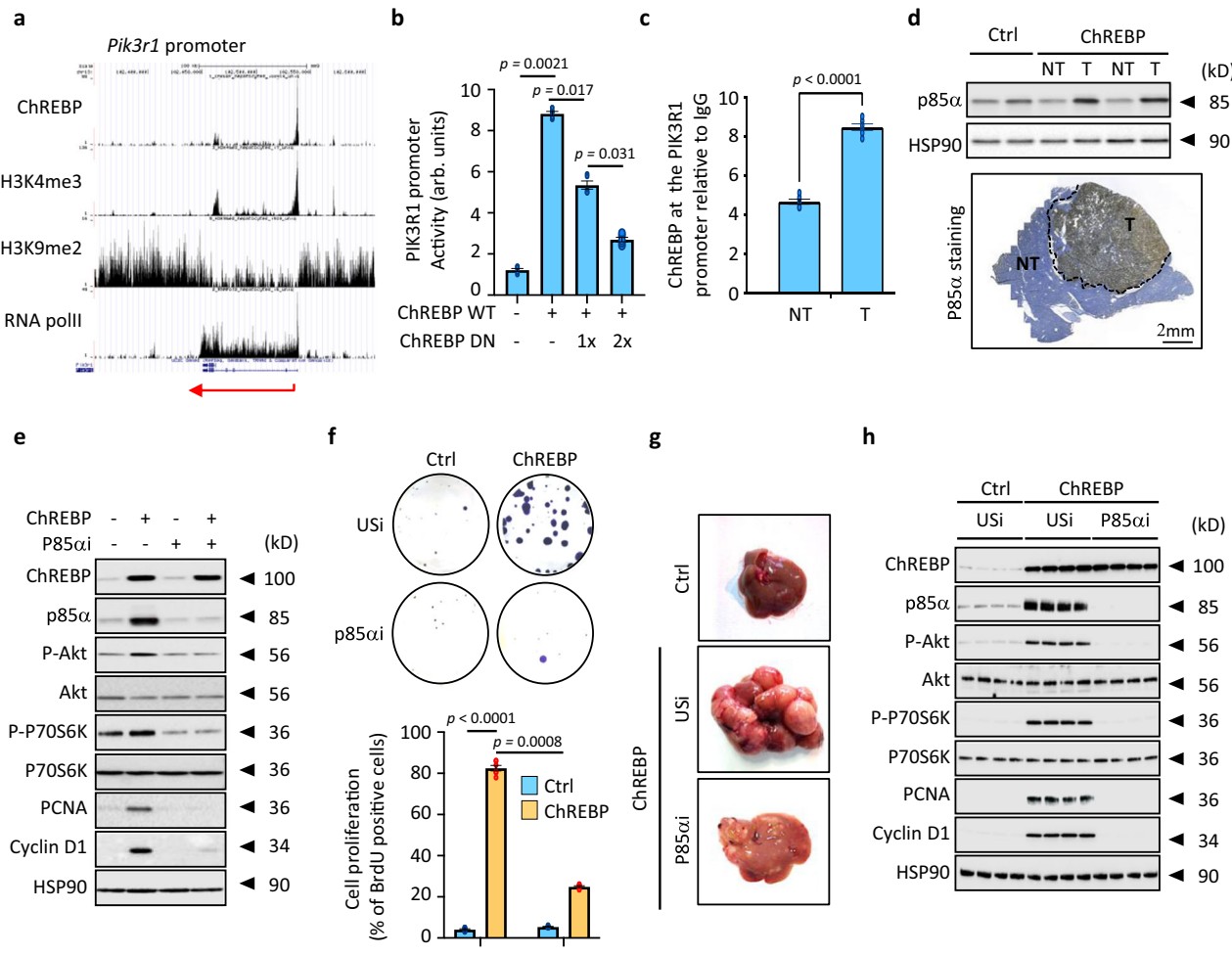

**Fig. 4 | p85α drives the stimulatory effects of ChREBP on PI3K/AKT signaling and hepatocyte proliferation. a** UCSC genome browser image illustrating normalized tag counts for ChREBP, H3K4me3, H3K9me2 and RNA polII at the *Pik3r1* gene promoter. **b** *Pik3r1* promoter activity in HEK 293 cells after ChREBP overexpression. A dominant negative form of ChREBP (DN ChREBP) was co-overexpressed to antagonize ChREBP action on *Pik3r1* promoter. *Pik3r1* promoter activity is relative to the RSV-β-galactosidase activity (arb. units = arbitrary unit) (*n* = 3 independent experiments). **c** ChIP experiments measuring ChREBP occupancy at the *Pik3r1* promoter in ChREBP tumors relative to IgG controls (*n* = 6 biologically independent *mice* per group). **d** (Top panel) Representative Western blot analysis of p85α expression in ChREBP tumors (*n* = 10 biologically independent *mice* per group). (Bottom panel) Representative staining of liver sections with p85α antibody from ChREBP tumors (*n* = 10 biologically independent *mice* per group). NT, non-tumoral tissue; T, tumors. Scale bar = 2 mm. **e, f** P85α was stably inhibited

in Huh7 cell line overexpressing ChREBP through Crispr/Cas9. **e** Representative Western blot analysis of PI3K/AKT signaling (*n* = 5). **f** (Top panel) Representative clonogenic assays shown (*n* = 8 independent experiments). (Bottom panel) Cell proliferation index determined by measuring the % of BrdU positive Huh7 cells (*n* = 5 independent experiments). **g, h** P85α was stably inhibited through Crispr/Cas9 in the liver of C57BL6/J *mice* overexpressing ChREBP using hydrodynamic injection. **g** Representative of tumor morphology is shown out of 10 *mice* per group (*n* = 10 biologically independent *mice* per group). **h** Representative Western blot of ChREBP and P85α expression levels in addition to PI3K/AKT signaling in tumors (*n* = 10 biologically independent *mice* per group). All error bars represent mean ± SEM. **b** Statistical analyses were determined by two-way ANOVA and Tukey's multiple-comparisons test. **c** Statistical analyses were determined by unpaired two-sided Student's *t* test. Source data are provided as a Source Data file.

progression of HCC at least in part *via* enhancing the PI3K/AKT signaling in the liver.

## Hexokinase 2 is a component of an amplification loop that connects glucose mediated ChREBP activation to increased PI3K/AKT signaling

Surprisingly, as the direct result of p85α knockdown, the transcriptional activity of ChREBP itself was however unexpectedly reduced (Fig. 5a and Supplementary Fig. 6a). Interestingly, this decrease in ChREBP activity was the result of its cytosolic retention (Fig. 5b). This led us to hypothesize the existence of a positive amplification feedback loop in which ChREBP-mediated p85α expression and subsequent activation of the PI3K/AKT signaling pathway could support ChREBP activation itself. ChREBP nuclear translocation is dependent

on the phosphorylation of glucose into glucose 6-phosphate (G6P)[27], a step mostly catalyzed in cancer cells by the hexokinase 2 (HK2), whose expression is under the direct control of the PI3K/AKT signaling pathway[28]. Because of increased PI3K/AKT signaling in ChREBP overexpressing hepatocytes, HK2 expression and activity were significantly enhanced (Fig. 5b, c). As a result, the concentration of G6P in ChREBP overexpressing hepatocytes increased (Fig. 5d), facilitating ChREBP nuclear translocation and activation (Fig. 5b). In contrast, when p85α expression was knocked down, ChREBP overexpression could no longer stimulate HK2 expression (Fig. 5b, c). As a result, G6P concentrations in ChREBP overexpressing livers were significantly lower (Fig. 5d), resulting in ChREBP cytosolic retention (Fig. 5b). Collectively, these results demonstrate that HK2 can connect and amplify both growth factors signaling with ChREBP-mediated glucose metabolic

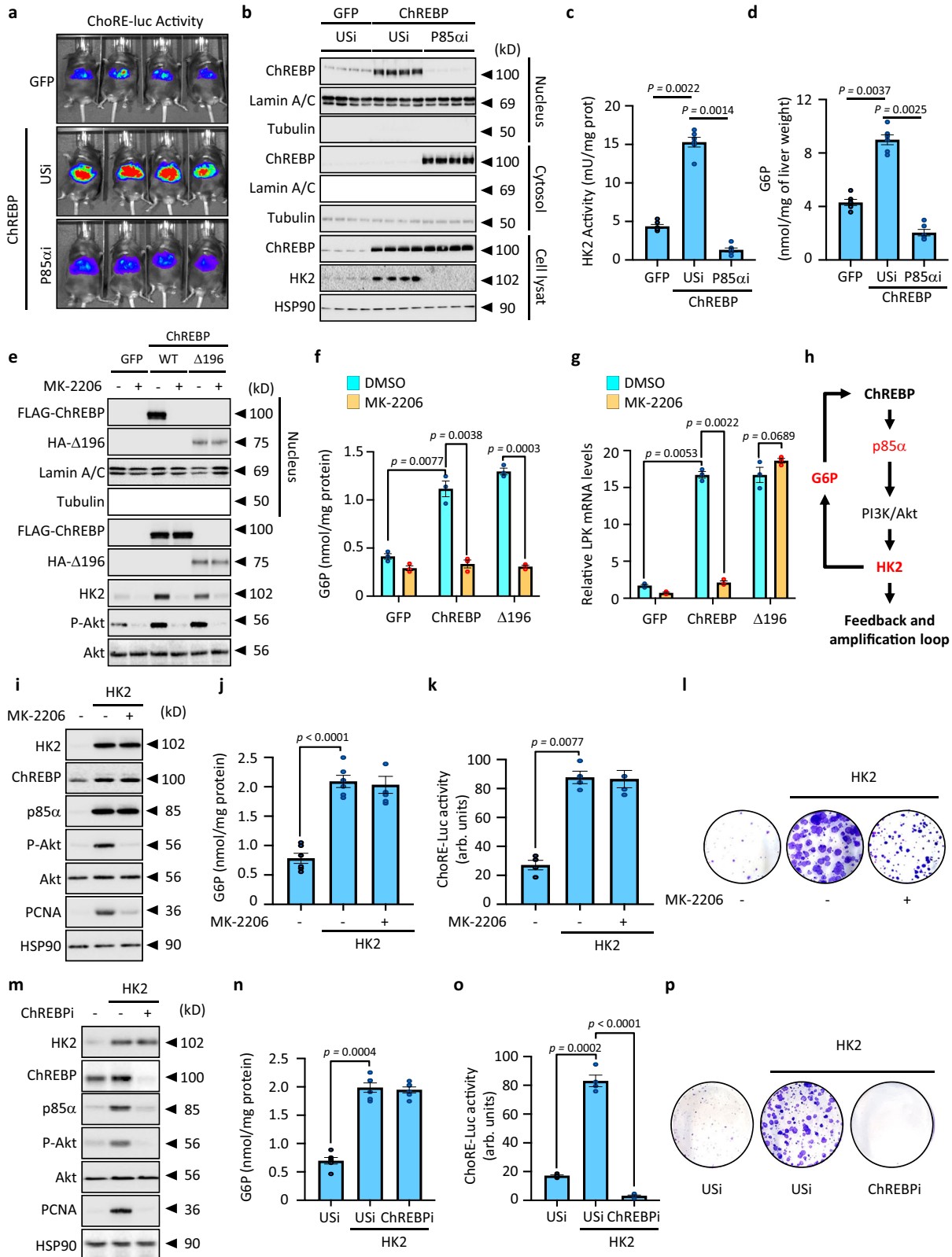

reprogramming to support cell proliferation and cancer development. To test this hypothesis further, we overexpressed in Huh7 cells a constitutive active form of ChREBP (Δ196), deleted from the first 196 amino acids of the protein that contains the ChREBP glucose sensing module (GSM)[29]. This isoform is constitutively nuclear and thus unaffected by HK2's phosphorylation of glucose into G6P. Overexpression of either FLAG tagged WT or HA tagged ChREBP Δ196 isoforms

increased PI3K/AKT signaling in comparison to GFP cells (Fig. 5e). As a result, HK2 activity increased in both cases, as evidenced by increased G6P production (Fig. 5f and Supplementary Fig. 6b). In this setting, both ChREBP isoforms were able to potentiate the expression of LPK, a bonafide ChREBP target gene (Fig. 5g). As previously observed, the AKT inhibitor MK-2206 significantly reduced PI3K/AKT signaling in ChREBP (WT or Δ196) overexpressing Huh7 cells (Fig. 5e).

**Fig. 5 | Hexokinase 2 is a part of an amplification loop linking glucose mediated ChREBP activation with enhanced PI3K/AKT signaling. a–d** C57BL6/J male *mice* were injected with either GFP or ChREBP overexpressing adenovirus. Simultaneously, P85α expression was also inhibited through adenoviral-mediated shRNA delivery. *Mice* were study 3 weeks later. **a** Representative in vivo bioluminescence imaging depicting ChREBP activity on ChREBP-regulated reporter construct (ChoRE-luc) (*n* = 6 biologically independent *mice* per group). **b** Representative Western blot analysis of ChREBP sub-cellular localization in response to P85α silencing (*n* = 6 biologically independent *mice* per group). **c** Measurement of HK2 activity (*n* = 6 biologically independent *mice* per group). **d** Measurement of G6P concentration (*n* = 6 biologically independent *mice* per group). **e–h** FLAG tagged WT or HA tagged Δ196 isoforms of ChREBP were overexpressed in Huh7 cells. Cells were then treated with 100 nM of MK-2206 for 24h. **e** Representative Western blot depicting ChREBP cellular localization and PI3K/AKT signaling in response to MK-2206 treatment (*n* = 3 independent experiments). **f** Measurement of G6P concentration (*n* = 3 independent experiments). **g** LPK expression relative to the TBP gene expression (*n* = 3 independent experiments). **h** Model of the amplification

loop linking glucose mediated ChREBP activation with enhanced PI3K/AKT signaling. **i–l** HK2 was overexpressed in Huh7 cells using HK2 expressing adenovirus. After HK2 overexpression (24 h), cells were then treated with 100 nM of MK-2206 for 24 h. **i** Representative Western blot showing the effect of HK2 expression on PI3K/AKT signaling (*n* = 6 independent experiments). **j** Measurement of G6P concentration in Huh7 cells (*n* = 6 independent experiments). **k** Measurement of ChREBP transcriptional activity on ChoRE-luc reporter construct (*n* = 6 independent experiments). **l** Representative clonogenic assays shown (*n* = 6 independent experiments). **m–p** ChREBP expression was inhibited in HK2 overexpressing Huh7 cells. **m** Representative Western blot depicting the activity of the PI3K/AKT signaling pathway in response to ChREBP silencing (*n* = 6 independent experiments). **n** Measurement of G6P concentration in Huh7 cells (*n* = 6 independent experiments). **o** Measurement of ChREBP transcriptional activity on ChoRE-luc reporter construct (*n* = 6 independent experiments). **p** Representative clonogenic assays shown (*n* = 6 independent experiments). All error bars represent mean ± SEM. All statistical analyses were made using two-way ANOVA and Tukey's multiple-comparisons test. Source data are provided as a Source Data file.

Consequently, MK-2206 treatment inhibited the ChREBP-mediated increase in HK2 expression and activity, as revealed by a decrease in G6P production in all conditions (Fig. 5f and Supplementary Fig. 6b). As a result, the WT form of ChREBP was no longer translocated into the nucleus of Huh7 cells and thus was unable to enhance LPK gene expression. In contrast however, ChREBP Δ196 isoform was insensitive to HK2 inhibition by MK-2206 treatment as revealed by its persistence in the nucleus and its capacity to stimulate LPK expression (Fig. 5e, g). These findings supported the hypothesis of a feedback amplification circuitry in which ChREBP-mediated p85α expression exacerbates PI3K/AKT signaling to drive HK2 expression, which then potentiates ChREBP activity to promote HCC development (Fig. 5h). To test this hypothesis, HK2 was finally overexpressed in Huh7 cells using an adenoviral strategy (Fig. 5i). HK2 overexpression increased G6P production compared to control cells, which increased ChREBP transcriptional activity (Fig. 5j, k). As a result, P85α expression was increased, which in turn potentiated PI3K/AKT signaling and cell proliferation (Fig. 5i, l). More importantly, although MK-2206 treatment significantly impaired p85α-mediated increase in PI3K/AKT signaling, ChREBP activity was still increased in response to HK2 overexpression (Fig. 5i–l). Consequently, MK-2206 treatment does not completely inhibit cell proliferation when compared to HK2 overexpressing cells because ChREBP activity is maintained in response to HK2 overexpression (Fig. 5l). Finally, ChREBP silencing in HK2-overexpressing Huh7 cells not only abolished HK2-mediated increase in PI3K/AKT signaling but also completely compromised cell proliferation (Fig. 5m–p). Overall, these findings show that ChREBP also independently contributes to cell proliferation, regardless of its role in controlling growth factor signaling.

## ChREBP rewires glucose metabolism to favor glycolysis over oxidative phosphorylation

Gene ontology analysis of our transcriptomic and ChIP-seq datasets revealed that ChREBP activation affected the expression of genes involved in glycolysis, PPP, lipid synthesis, as well as glutamine and pyrimidine metabolism, indicating a second function of ChREBP in controlling HCC development (Supplementary Fig. 7a). These findings suggest that ChREBP activation of the PI3K/AKT signaling promotes a high metabolic demand, which drives HCC development. To better understand its role in reprogramming cellular energy metabolism, ChREBP was stably overexpressed in SNU449 and SNU475 *human* HCC cell lines, resulting in increased cell proliferation (Fig. 6a, b and Supplementary Fig. 8a, b). As a direct result of ChREBP overexpression, glucose uptake was significantly enhanced in both cell lines (Fig. 6c and supplementary Fig. 8c). The intracellular fate of [$^{13}C_6$] glucose was next investigated through stable isotope tracing approaches to evaluate how glucose was metabolized upon ChREBP-mediated activation

of glucose metabolism. Both parental and ChREBP overexpressing SNU449 and SNU475 cells were labeled for 6 h with 11 mM of [$^{13}C_6$] glucose. Cellular extracts were then subjected to tandem mass spectrometry (LC-MS/MS) analysis to identify significant changes in label incorporation into glucose-derived metabolites[30]. In this setting, major fates of glucose-derived pyruvate in HCC cells include reduction to lactate by lactate dehydrogenase (Ldha), transamination to alanine by alanine transaminase 1 (Gpt1), or oxidation in the TCA cycle (Supplementary Fig. 7b). ChREBP overexpressing SNU449 or SNU475 HCC cells displayed a high glucose-to-lactate (m+3) and high glucose-to-alanine (m+3) conversion, consistent with a metabolic shift over glycolysis (Warburg effect) (Fig. 6d and Supplementary Fig. 8d). Supporting these results, relative to parental cells, both SNU449 and SNU475 cells overexpressing ChREBP displayed a characteristic increase in basal glycolysis and glycolytic capacity, as demonstrated by a 2-fold increase in their extracellular acidification rate (ECAR) (Fig. 6e, Supplementary Figs. 7c and 8e). Because glycolytic rate was enhanced upon ChREBP overexpression, the fate of pyruvate, the end-product of glycolysis, into the TCA cycle was next investigated. Pyruvate (m+3) can be either (i) decarboxylated into acetyl-CoA by the PDH complex, thereby entering the TCA cycle as citrate, that contains two labeled carbons (m+2), or (ii) carboxylated by pyruvate carboxylase (PC), that entered the TCA cycle as oxaloacetate, that contains three labeled carbons (m+3) when combined with acetyl-CoA (Supplementary Fig. 7b). ChREBP overexpressing SNU449 and SNU475 cells displayed a prominent PDH and PC activity signature, as evidenced by the enrichment of m+2, m+3 and m+5 isotopologues of citrate (Fig. 6f, g and Supplementary Fig. 8f, g). However, despite this apparent increase in PDH and PC activity, [$^{13}C_6$] glucose labeling into m+2 TCA cycle intermediates (succinate, fumarate and malate) was significantly reduced in ChREBP overexpressing cells, suggesting reduced TCA cycle activity upon ChREBP activation (Fig. 6f and Supplementary Fig. 8f). Consistent with the lower TCA cycle activity, basal and ATP-linked oxygen consumption rates (OCR) were decreased in ChREBP overexpressing cells (Fig. 6h, Supplementary Figs. 7d and e and Supplementary Fig. 8h). Finally, we converted ECAR and OCR measurements into glycolytic and oxidative ATP production rates using Mookerjee and Brand's methodology[31], to assess the contribution of glycolysis and oxidative phosphorylation (OXPHOS) to overall ChREBP overexpression cell bioenergetics. This allowed us to compare the contributions of glycolysis and OXPHOS to ATP production while controlling for glycolysis and TCA cycle metabolism contributions to ECAR. When compared to parental cells, ChREBP overexpressing SNU449 and SNU475 cells produced significantly more ATP (Fig. 6i and Supplementary Fig. 8i). More importantly, ChREBP overexpressing cells derived most of their ATP (approximately 60%) from glycolysis, as opposed to parental cells, which relied primarily on oxidative

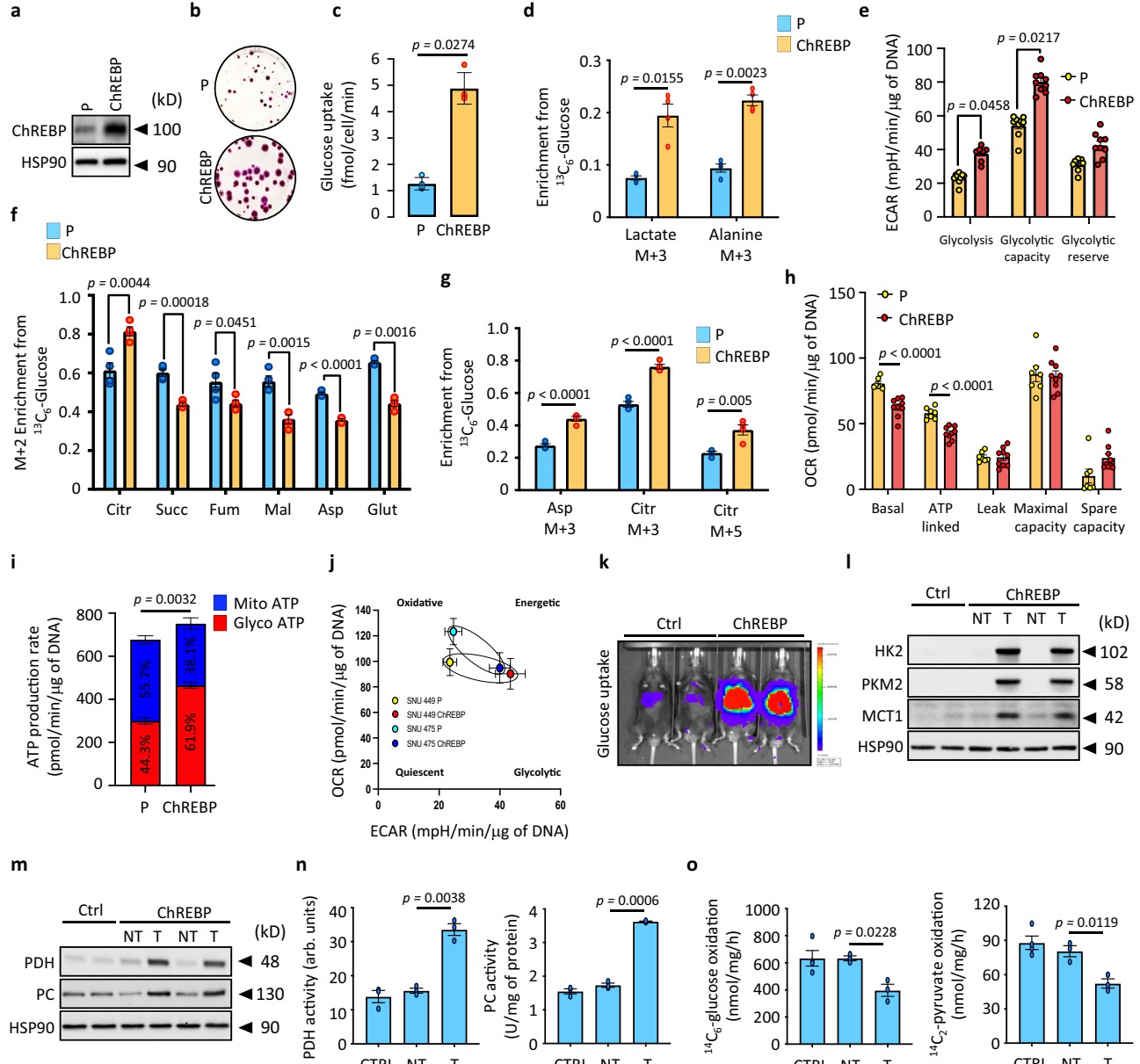

**Fig. 6 | ChREBP overexpression reroutes glucose metabolism away from oxidation to support glycolysis.** ChREBP was stably overexpressed using the Crispr/Cas9 technology in SNU449 hepatoma cell line. **a** Representative Western blot illustrating stable ChREBP overexpression. **b** Representative clonogenic assays shown (n = 3 independent experiments). **c** Rate of glucose uptake (n = 5 independent experiments). **d** Enrichment in (m+3) lactate and (m+3) alanine from $^{13}C_6$-glucose in response to ChREBP overexpression (n = 4 independent experiments). **e** Graph showing glycolysis, glycolytic capacity, and glycolytic reserve of parental and ChREBP overexpressing SNU449 (n = 9 independent experiments). Representative curve can be found in supplementary Fig. 7c. **f** (m+2) enrichment in citrate (citr), succinate (succ), fumarate (fum), malate (mal), aspartate (asp) and glutamate (glut) from $^{13}C_6$-glucose in response to ChREBP overexpression (n = 4 independent experiments). **g** Enrichment in (m+3) aspartate, (m+3) and (m+5) citrate from $^{13}C_6$-glucose in response to ChREBP overexpression (n = 4 independent experiments). **h** Graph showing basal OCR, proton leakage, maximal respiration, spare capacity, and ATP production of parental and ChREBP overexpressing SNU449 (n = 9 independent experiments). Representative profile after mitochondrial stress assay showing the OCR of these SNU449 cells can be found in Supplementary Fig. 7d.

**i** ATP production rate from glycolysis or oxidative phosphorylation in parental or ChREBP overexpressing SNU449 (n = 9 independent experiments). **j** Energy map charting basal OCR versus basal glycolysis values (n = 9 independent experiments). **k–o** ChREBP was stably overexpressed in liver of C57Bl6/J *mice* using the SB transposon system as previously described. **k** Representative image of liver glucose uptake of ChREBP overexpressing *mice* with BiGluc probe as described in[83] (n = 5 biologically independent *mice* per group). **l** Representative Western blot analysis of glycolytic enzymes (n = 10 biologically independent *mice* per group).

**m** Representative Western blot analysis of PDH and PC protein content in ChREBP tumors (n = 10 biologically independent *mice* per group). **n** Measurement of PDH and PC activity (n = 3 biologically independent *mice* per group). **o** Glucose and pyruvate oxidation rate determined by measuring the production of $^{14}CO_2$ from $^{14}C_6$-glucose or $^{14}C$-(2)-pyruvate for 4 h (n = 4 biologically independent *mice* per group). All error bars represent mean ± SEM. **c–i** Statistical analyses were determined by unpaired two-sided Student's *t* test. **n**, **o** All statistical analyses were made using two-way ANOVA and Tukey's multiple-comparisons test. Source data are provided as a Source Data file.

phosphorylation (Fig. 6i and Supplementary Fig. 8i). Thus, proliferating ChREBP overexpressing SNU449 and SNU475 cells have a distinct bioenergetic profile that is characterized by increased glycolytic activity with decreased oxidative ATP production, higher bioenergetic capacity, and increased flexibility in switching between ATP production pathways (Fig. 6j). Finally, all these findings were validated in vivo in ChREBP overexpression tumors. As previously observed in vitro, ChREBP over-expressing *mice* elicited an increase in liver glucose uptake compared to control *mice* (Fig. 6k). In accordance with these observations, the expression of key HCC glucose transporters (*slc2a1, slc2a2* and *slc2a3*), as well as of key glycolytic enzymes (*hk2, gpi, pfk, aldoa, pkm2*) was strongly up regulated in ChREBP overexpressing tumors (Fig. 6l and Supplementary Fig. 9a). In this context, the expression of MCT1 (monocarboxylic Acid Transporter 1, known as *slc16a1*), which is the main lactate transporter across the plasma membrane, was increased in ChREBP overexpressing tumors (Fig. 6l and Supplementary Fig. 9a). This further confirms that the extracellular acidification observed in response to ChREBP activation is primarily due to proton symport with lactate excretion from glucose. Importantly, despite higher glucose uptake and subsequent activation of glycolysis, concentrations of all glycolytic intermediates, determined by mass spectrometry, were reduced in these ChREBP overexpressing tumors (Supplementary Fig. 9b), reinforcing our observations that these substrates are rapidly consumed in response to ChREBP-mediated activation of glucose metabolism. Despite increased expression and activity of PDH and PC in ChREBP tumors (Fig. 6m, n), mitochondrial oxidation of $^{14}C_6$-labeled glucose or $^{14}C_2$-labeled pyruvate was reduced (Fig. 6o). These findings support the idea that glucose metabolism in ChREBP tumors is rerouted away from oxidation to drive continued metabolic demand and tumor growth.

### ChREBP contributes to nucleotide biosynthesis by rewiring glucose metabolic fluxes into the Pentose Phosphate Pathway (PPP)

Based on our bioinformatic analysis, such specific anabolic pathways may include de novo nucleotide and fatty acid synthesis. However, as previously observed for glycolytic intermediates, the concentration of main intermediates of the PPP, determined by mass spectrometry analysis, was reduced in ChREBP overexpressing tumors (Supplementary Fig. 9b). This suggests again that glucose was rapidly consumed in response to ChREBP activation into the PPP to sustain de novo nucleotides biosynthesis. Thus, to determine the contribution of ChREBP to nucleotide synthesis from glucose (visualization of metabolic flux through the PPP), we perform a short pulse (30 min) of $[^{13}C_6]$-glucose along with 6-aminonicotinamide (6-AN: a competitive inhibitor of 6-phosphogluconate dehydrogenase) in parental or ChREBP overexpressing SNU449 or SNU475 (Supplementary Fig. 9c). ChREBP activation led to a striking increase in 6-phosphogluconate levels and label incorporation (m+6) from $[^{13}C_6]$ glucose in the presence of 6-AN, revealing high ChREBP-driven flux through this pathway (Fig. 7a and Supplementary Fig. 8j). The isotopologue distribution of ribose 5-phosphate showed that most of the ChREBP-driven synthesis of ribose 5-phosphate implicated the oxidative arm (m+5) of the PPP (Fig. 7b and Supplementary Fig. 8k). Importantly, the oxidative (m+5) contribution of ribose 5-phosphate was sensitive to 6-AN inhibition, allowing the rerouting of glycolytic carbons through the reversed non-oxidative arm of the PPP (Fig. 7b and Supplementary Fig. 8k). Consequently, there was a significant proportion of ribose 5-phosphate that was also labeled (m+3) (Fig. 7c and Supplementary Fig. 8l). As expected however, the (m+3) contribution to ribose 5-phosphate synthesis was not sensitive to 6-AN treatment, indicating that it arose through the non-oxidative arm of the pathway from glyceraldehyde 3-phosphate (G3P). These findings suggest that the high demand for R5P caused by ChREBP activation rerouted glycolytic carbon through both the oxidative and non-oxidative arms of the PPP, favoring de novo nucleotide

synthesis. Supporting these results, de novo nucleotide synthesis from $^{14}C_6$-labeled glucose was significantly potentiated in both SNU449 and SNU475 cells overexpressing ChREBP, effect that was largely prevented upon 6-AN treatment (Fig. 7d and Supplementary Fig. 8m). As a result, 6-AN treatment abolished ChREBP-mediated increase in SNU449 and SNU475 cell proliferation (Fig. 7e, f and Supplementary Fig. 8n, o). More importantly, dNTPs supplementation was able to rescue ChREBP-mediated increase in cell proliferation after 6-AN treatment, supporting the importance of the PPP and nucleotide synthesis in ChREBP pro-proliferative effects (Fig. 7f and Supplementary Fig. 8o). Finally, supporting these results, the expression of key enzymes involved in the oxidative and non-oxidative arms of the PPP, including G6PDH, PGD, TKT and RPIA, was increased in ChREBP overexpressing tumors in vivo (Fig. 7g and Supplementary Fig. 9a). Therefore, as observed in SNU449 and SNU475 HCC cells, de novo synthesis of nucleotides from $^{14}C_6$-labeled glucose was enhanced in ChREBP overexpressing tumors compared to non-tumoral adjacent tissues (Fig. 7h). At the molecular level, ChIP-seq analysis further revealed that ChREBP was functionally recruited to the promoter of the *g6pd, pgd, tkt* and *rpia* (Supplementary Fig. 9d). Accordingly, we observed that the recruitment of ChREBP to the promoter of these PPP genes was indeed enhanced in ChREBP tumors to simulate their expression (Fig. 7i). Finally, relative NADPH/NADP ratio was increased in ChREBP overexpressing tumors, overall supporting this enhanced activity of the PPP in response to ChREBP activation (Fig. 7j).

### ChREBP promotes glucose metabolic rewiring into de novo fatty acid synthesis to drive continued metabolic demand and tumor growth

In addition to de novo nucleotides synthesis, increased $^{14}C_6$-labeled glucose flux towards de novo lipid synthesis was also observed in ChREBP tumors (Fig. 7k). Accordingly, expression of key lipogenic genes (*acaca, fasn, scd1*) in parallel to ChREBP recruitment to their respective promoter were increased in ChREBP tumors (Fig. 7l, m and Supplementary Fig. 9a, e). Additional lipidomic analyses showed that ChREBP tumors displayed a distinct lipid profile compared to corresponding non tumoral tissues (Supplementary Fig. 10). ChREBP-mediated changes were characterized by a significant increase in the ratio of monounsaturated fatty acid (MUFA) in triglycerides (TGs) and all species of phospholipids (PLs) (Supplementary Fig. 10a–f), consistent with enhanced de novo lipogenesis and SCD1-associated desaturation of lipid species (Fig. 7l, n). Positive correlation of MUFA-PLs with lipogenic genes expression and levels of metabolites involved in PLs synthesis, such as CDP-choline (Supplementary Fig. 10f), further suggested that ChREBP-driven proliferation might depend on de novo lipogenesis and SCD-mediated desaturation of fatty acids. Thus, by increasing the pool of MUFAs and MUFA-PLs, ChREBP may contribute to membrane lipid diversity during HCC development. Consistently, crispr/cas9-mediated inhibition of FAS expression impaired ChREBP-dependent increase of de novo fatty acid synthesis from $^{14}C_6$-labeled glucose and ChREBP pro-proliferative effects in both SNU449 and SNU475 cells (Fig. 7o–r and Supplementary Fig. 8p–s). Of note, oleate supplementation was able to fully rescue the inhibition of ChREBP-driven proliferation upon FAS knockdown (Fig. 7o–r and Supplementary Fig. 8p–s). Finally, the importance of ChREBP in mediating glucose metabolic rewiring was explored in *human*s by assessing 10 independent HCC microarray datasets previously used in this study (Supplementary Table 1). In *human* hepatic tumors, a total of 285 genes involved in metabolic pathways such as glycolysis, PPP, nucleotides synthesis and lipid metabolism were upregulated compared to non-tumoral adjacent tissues and overlapped with newly identified ChREBP-regulated genes in our study (Supplementary Data 8). Therefore, ChREBP-driven alterations in glycolysis, PPP and de novo lipogenesis are consistent across our preclinical models and clinical cohorts and support a key role of

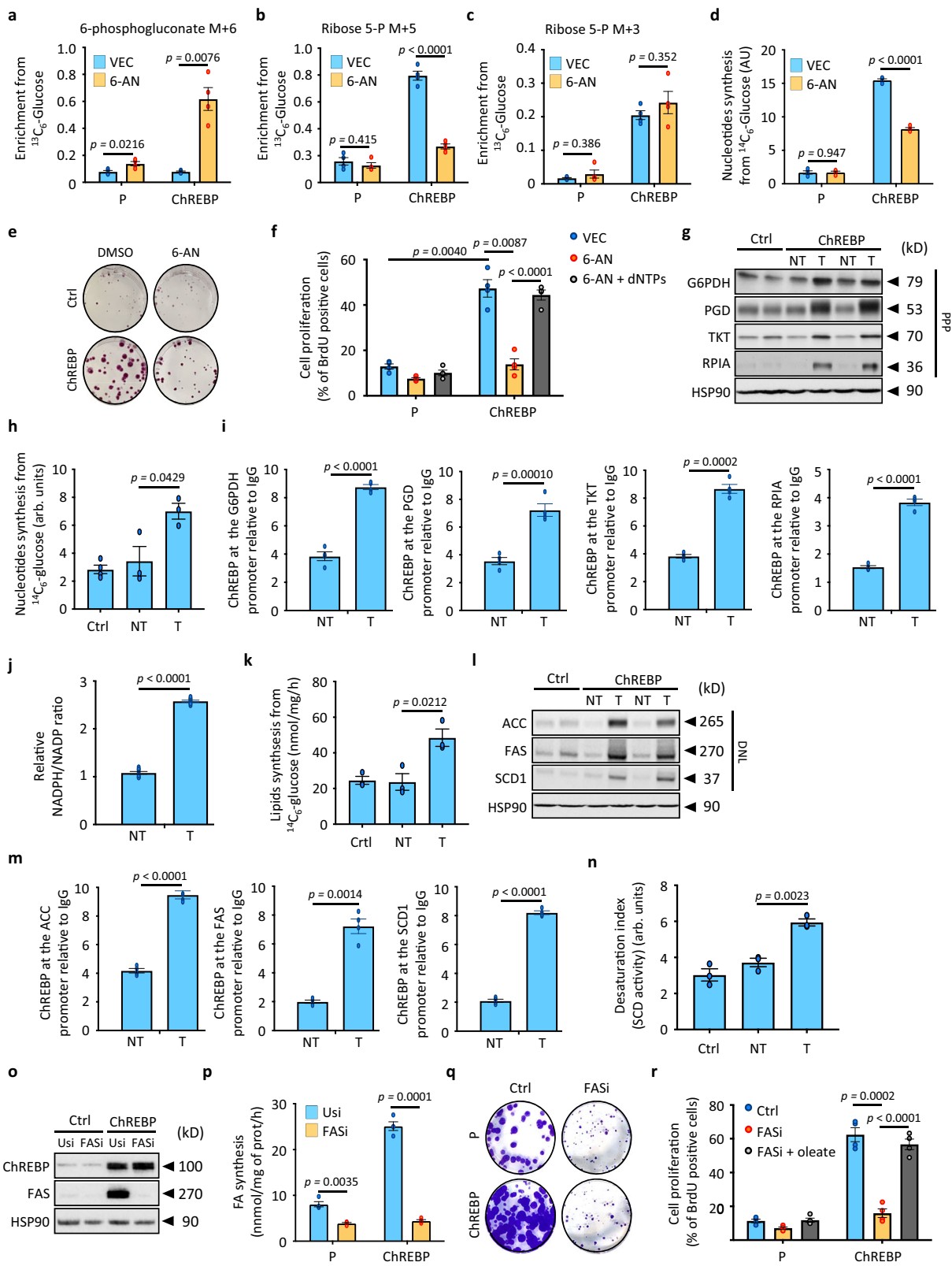

ChREBP in rewiring glucose metabolism in the context of *human* liver carcinogenesis.

## ChREBP favors tumor growth by redirecting glutamine metabolism into aspartate production

For the first time, gene ontology analyses revealed that genes involved in the regulation of the "non-essential" amino acid glutamine (Gln) were differentially affected in ChREBP tumors, implying that ChREBP's function may not be limited to glucose metabolic rewiring (Supplementary Fig. 7a). Accordingly, the expression of key genes involved in Gln handling and processing such as *Scl1a5*, *gls1* and *got2* was increased in ChREBP tumors (Fig. 8a and Supplementary Fig. 11a) along with the concentration of key intermediates of Gln metabolism including N-acetylglutamine,

**Fig. 7 | Metabolic rewiring of glucose metabolism into the PPP and de novo lipogenesis participates to ChREBP pro-proliferative effects. a–c** Parental and ChREBP overexpressing SNU449 cells were treated with 6-AN (6-aminonicotinamide, 40 μM) for 24 h. Cells were then incubated for 30 min with 11 mM of $^{13}C_6$-glucose. **a** Enrichment in (m+6) 6-phosphogluconate from $^{13}C_6$-glucose ($n = 4$ independent experiments). **b** Enrichment in (m+5) ribose 5-phopshate from $^{13}C_6$-glucose in response to ChREBP overexpression ($n = 4$ independent experiments). **c** Enrichment in (m+3) ribose 5-phopshate from $^{13}C_6$-glucose in response to ChREBP overexpression ($n = 4$ independent experiments). **d** De novo nucleotide synthesis from parental and ChREBP overexpressing SNU449 cells incubated 6 h with 11 mM of $^{14}C_6$-labeled glucose ($n = 4$ independent experiments). **e** Effect of 6-AN treatment (40 μM, 24 h) on ChREBP-mediated increase in hepatocyte proliferation was studied in SNU449 cells. Representative clonogenic assays shown ($n = 7$ independent experiments). **f** Effect of 6-AN treatment (40 μM, 24 h) and dNTPs rescue (100 μM each) on ChREBP-mediated increase in cell proliferation. Cell proliferation index was determined by measuring the % of BrdU positive cells ($n = 4$ independent experiments). **g** Representative Western blot analysis of proteins involved in PPP pathway ($n = 10$ biologically independent *mice* per group). **h** De novo nucleotide synthesis from $^{14}C_6$-labeled glucose in ChREBP tumors ($n = 3$ biologically independent *mice* per group). **i** ChIP experiments measuring ChREBP occupancy at the G6PDH, PGD, TKT and RPIA promoters in ChREBP tumors relative to IgG controls ($n = 4$ biologically independent *mice* per group). **j** Relative NADPH/NADP ratio in ChREBP overexpressing tumors ($n = 6$ biologically independent *mice* per group). **k** De novo lipid synthesis from $^{14}C_6$-labeled glucose in ChREBP tumors ($n = 3$ biologically independent *mice* per group). **l** Representative Western blot analysis of proteins involved in de novo lipogenic pathways ($n = 10$ biologically independent *mice* per group). **m** ChIP experiments measuring ChREBP occupancy at the ACC, FAS and SCD1 promoters in ChREBP tumors relative to IgG controls ($n = 4$ biologically independent *mice* per group). **n** Measurement of SCD1 activity in ChREBP tumors ($n = 3$ biologically independent *mice* per group). **o–r** FAS expression was knockdown by Crispr/Cas9 in ChREBP overexpressing SNU449 cells (FASi). **o** Representative Western blot showing FAS deletion in ChREBP overexpressing SNU449 ($n = 3$ independent experiments). **p** De novo lipid synthesis from $^{14}C_6$-labeled glucose. SNU449 cells were incubated 6 h with 11 mM of $^{14}C_6$-labeled glucose ($n = 3$ independent experiments). **q** Representative clonogenic assays shown after FAS silencing ($n = 3$ independent experiments). **r** Effect of FAS silencing and oleate supplementation (50 μM) on ChREBP-mediated increase in cell proliferation. Cell proliferation index was determined by measuring the % of BrdU positive cells ($n = 3$ independent experiments). All error bars represent mean ± SEM. **a–d, f, h, k, n, p, r** Statistical analyses were made using two-way ANOVA and Tukey's multiple-comparisons test. **i, j** and **m** Statistical analyses were determined by unpaired two-sided Student's *t* test. Source data are provided as a Source Data file.

N-acetylglutamate or aspartate (Supplementary Fig. 11b). To further characterize ChREBP-induced changes in glutamine metabolism during cell proliferation, both SNU449 and SNU475 cells overexpressing ChREBP were labeled with 4 mM of $^{13}C_5$-glutamine for 6h (Supplementary Fig. 11c). Comparative LC-MS/MS analyses of related metabolites in both cell lines revealed that ChREBP overexpressing cells had a higher percentage of intracellular m+5 isotopologue-labeled glutamate (Glut) than parental cells (Fig. 8b and Supplementary Fig. 11d). Consistent with ChREBP-activated Gln conversion to Glut by GLS to support HCC cell growth, Gln deprivation or 6-Diazo-5-oxo-L-norleucine (DON)-induced GLS inhibition severely impaired ChREBP-driven cell proliferation (Fig. 8c and Supplementary Fig. 11e). However, Glut supplementation fully rescued ChREBP-mediated cell growth in Gln-free media or upon DON treatment. In cancer cells, conversion of Glut into α-ketoglutarate (α-KG) relies on glutamate dehydrogenase (GLUD1), glutamic-pyruvic transaminase 2 (GPT2) or glutamic-oxaloacetic transaminase 2 (GOT2) (Supplementary Fig. 11f). Typically, $^{13}C_5$-glutamine oxidative metabolism would predictably generate (m+4) isotopologues of succinate, malate, aspartate, and citrate owing to the incorporation of four $^{13}C$ atoms in each of these species (Supplementary Fig. 11c). In contrast, reductive carboxylation would predictably yield (m+5) forms of citrate and (m+3) forms of aspartate (Supplementary Fig. 11c). Thus, the relative contribution of oxidative metabolism versus reductive carboxylation arising from gln-glut-aKG flux can be confidently quantified by determining the isotopologue distribution and abundance of $^{13}C$ enrichment in TCA cycle intermediates. In SNU449 and SNU475 cells, ChREBP activation elicited a reduction of Gln oxidation, as illustrated by decreased enrichments in m+4-labeled isotopologues of succinate, fumarate, malate, aspartate, and citrate (Fig. 8b and Supplementary Fig. 11d). Accordingly, $^{14}C_5$-labeled Gln oxidation was also reduced in ChREBP overexpressing tumors in vivo (Supplementary Fig. 11g). By contrast, both ChREBP overexpressing cells displayed increased levels of m+5-labeled citrate and m+3-labeled aspartate (Fig. 8d and Supplementary Fig. 11h). In addition, de novo lipogenesis from $^{14}C_5$-labeled Gln was also increased in ChREBP tumors, suggesting a ChREBP-dependent raise in glutamine reductive carboxylation (Supplementary Fig. 11i). Even though many cancer cells rely on GLUD1-mediated Glut deamination to replenish the Krebs cycle, epigallocatechin gallate (EGCG) (50 μM), a GLUD1 inhibitor, had surprisingly no effect on ChREBP-mediated proliferation of SNU449 and SNU475 cells (Fig. 8e and Supplementary Fig. 11j). The cell permeable dimethyl α-KG (DMK, 6 mM) neither restored

growth upon Gln deprivation, indicating that ChREBP-driven Gln metabolism in HCC cells differs from canonical models and might rely on transaminases activity. To test this hypothesis, SNU449 and SNU475 cells overexpressing ChREBP were treated with aminooxyacetate (AOA), a pan transaminase inhibitor. AOA significantly dampened ChREBP-mediated cell growth, and supplementation with aspartate (Asp), but not alanine (Ala), respectively the output of GOT2 and GPT, was able to rescue this inhibition (Fig. 8f and Supplementary Fig. 11k). ChIP-seq analysis finally revealed that ChREBP is functionally recruited to the promoter of GLS1 and GOT2, and additional ChIP experiments validated that enhanced ChREBP recruitment on GLS1 and GOT2 promoters promoted the expression of GSL1 and GOT2 in ChREBP tumors (Fig. 8a, g and Supplementary Fig. 11a, l). Overall, these results unveiled that ChREBP, by directly regulating GLS1 and GOT2 expression, rerouted Gln metabolism to fuel Asp production and thus support high proliferation rate of HCC cancer cells (Supplementary Fig. 11m).

## ChREBP activation coordinates the convergence of Gln and glucose metabolisms to fuel de novo pyrimidine synthesis

Gln not only provides carbon sources for the TCA cycle but also supports anabolic processes. Therefore, since glutamine and glutamine-derived aspartate are required for nucleotides biosynthesis, we reasoned that the GOT2-mediated conversion of Glut into Asp, in response to ChREBP activation, may generate nucleotides through de novo pyrimidine biosynthesis, Asp being the carbon donor (Supplementary Fig. 12a). Interestingly, the expression of key enzymes involved in de novo pyrimidine synthesis such as PRPS2, UMPS, CTPS1, was increased in response to ChREBP overexpression (Fig. 8h, i). At the molecular level, ChIP-seq experiment and ChIP-qPCR validation revealed that ChREBP was functionally recruited to the promoter of *umps* and *ctps1* (Fig. 8j and Supplementary Fig. 12b). Furthermore, when compared to non-tumoral adjacent tissues, ChREBP occupancy on these two promoters was increased in ChREBP tumors (Fig. 8j). Accordingly, the abundance of key intermediates of de novo pyrimidine synthesis was increased in ChREBP tumors (Supplementary Fig. 12c). In addition, the abundance of 5-phosphoribosyl-1-pyrophosphate (PRPP), a metabolite linking the PPP with the upstream steps of pyrimidine synthesis, was also increased in ChREBP tumors (Supplementary Fig. 12c). As a result, de novo nucleotide synthesis from $^{14}C_5$-labeled Gln or $^{14}C_4$-labeled Asp was enhanced in these tumors (Fig. 8k). Consistently, dCTP and dTTP pyrimidine pools, measured by HPLC, were increased in ChREBP

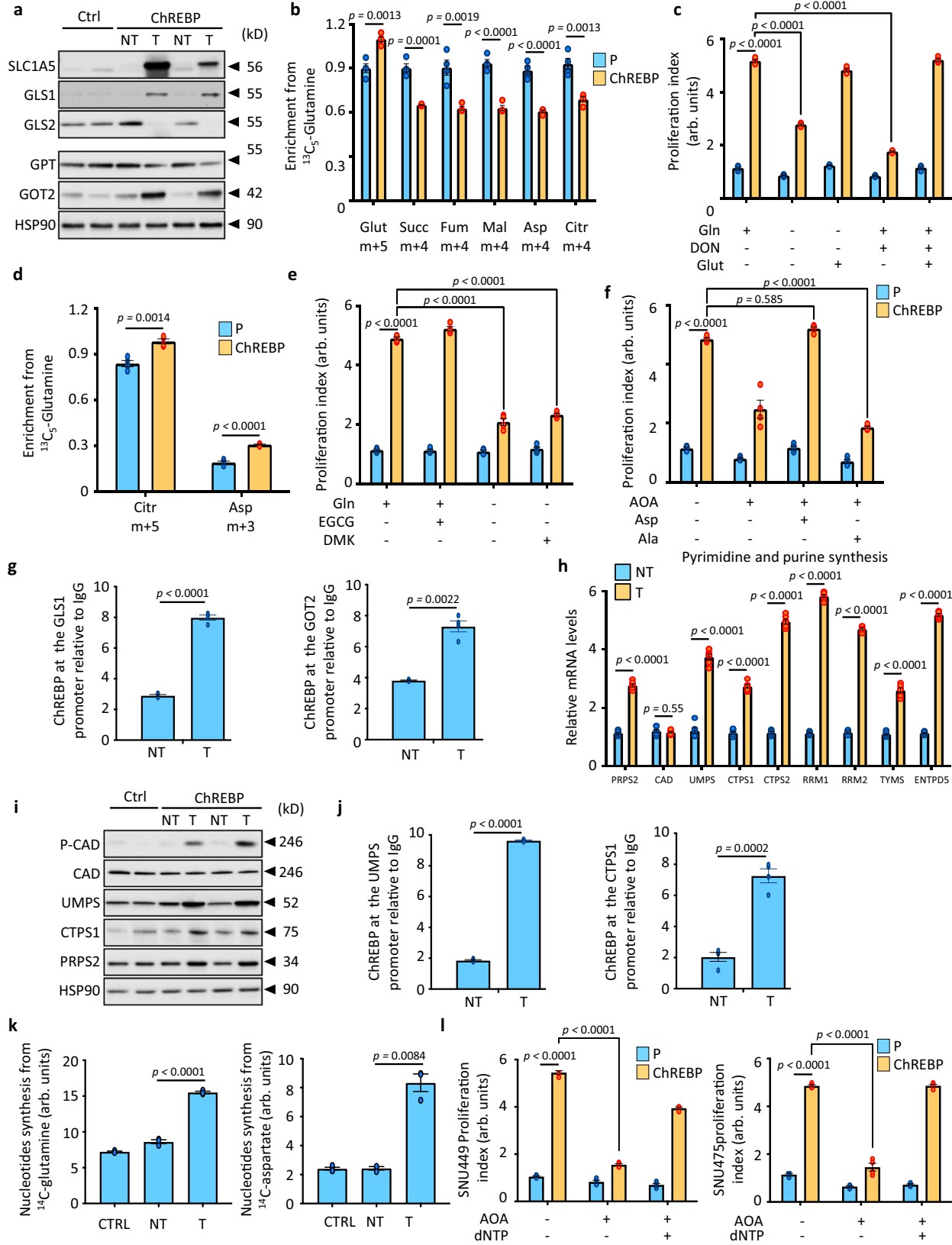

tumors (Supplementary Fig. 12d). The importance of pyrimidine synthesis in ChREBP action was also reinforced by the fact that dNTPs supplementation was able to rescue ChREBP-mediated increased in cell proliferation after AOA treatment in both SNU449 and SNU475 HCC cells (Fig. 8l). Finally, the expression of these newly identified ChREBP target genes (*gls1, got2, umps* and *ctps1*) were upregulated

along with ChREBP across the 10 *human* datasets used in this study (Supplementary Data 8). Altogether, these data demonstrate that alterations in glutamine metabolism and pyrimidine synthesis are consistent across HCC clinical cohorts and strengthen the role of ChREBP in rewiring glutamine metabolism during *human* liver carcinogenesis (Supplementary Fig. 11e).

**Fig. 8 | ChREBP favors tumor growth by channeling glutamine metabolism into de novo pyrimidine synthesis. a** Representative Western blot analysis of proteins involved in Gln processing ($n = 10$ biologically independent *mice* per group). **b** Enrichment in (m+5) glutamate and (m+4) succinate, (m+4) fumarate, (m+4) malate, (m+4) aspartate and (m+4) citrate from $^{13}C_5$-glutamine in response to ChREBP overexpression in SNU449 cells ($n = 4$ independent experiments). **c** Proliferation index of SNU449 cells overexpressing ChREBP after Gln deprivation or GLS inhibition ($n = 4$ independent experiments). Glutamate (Glu) 4 mM. **d** Enrichment in (m+3) citrate and (m+3) aspartate from $^{13}C_5$-glutamine in response to ChREBP overexpression ($n = 4$ independent experiments). **e** Proliferation index of SNU449 cells overexpressing ChREBP after GLUD1 inhibition ($n = 4$ independent experiments). Dimethyl-αKG (DMK) 2 mM. **f** Proliferation index of SNU449 cells overexpressing ChREBP after AOA treatment ($n = 4$ independent experiments). Aspartate (Asp) or Alanine (Ala) (0.1 mM). **g** ChIP experiments measuring ChREBP occupancy at the GLS1 and GOT2 promoters relative to IgG controls in ChREBP tumors ($n = 3$ biologically independent *mice* per group). **h** Expression of genes involved in purine and pyrimidine synthesis relative to the TBP gene expression ($n = 12$ biologically independent *mice* per group). **i** Representative Western blot analysis of proteins involved in de novo pyrimidine synthesis ($n = 10$ biologically independent *mice* per group). **j** ChIP experiments measuring ChREBP occupancy at the UMPS and CTPS1 promoters in ChREBP tumors relative to IgG controls ($n = 4$ biologically independent *mice* per group). **k** Measurement of de novo nucleotide synthesis from $^{14}C_5$-labeled Gln and $^{14}C_4$-labeled Asp in ChREBP tumors ($n = 3$ independent experiments). **l** Proliferation index of SNU449 and SNU475 cells overexpressing ChREBP after dNTPs supplementation (100 mM each) and AOA treatment ($n = 3$ independent experiments). All error bars represent mean ± SEM. **b**, **d**, **g**, **h**, **j** Statistical analyses were determined by unpaired two-sided Student's *t* test. **c**, **e**, **f**, **k**, **i** Statistical analyses were made using two-way ANOVA and Tukey's multiple-comparisons test. Source data are provided as a Source Data file.

## Targeting ChREBP represents a novel and promising strategy for HCC treatment

Because these findings show that ChREBP functions as a molecular link between growth signals and acute control over HCC cell energy metabolism, we next addressed the importance of ChREBP as a potential target for pharmacological intervention. To answer this question, we assessed the correlation between ChREBP expression with deregulation of the canonical PI3K, glycolysis, PPP, lipogenesis, glutaminolysis and pyrimidine synthesis pathways in the TCGA - LIHC dataset. Alterations in these canonical pathways were defined by the genetic modifications (activating or inactivating events) of their pathway members. If one or more genes in these pathways contained a recurrent or known driver alteration, the tumor was considered as affected. Consistently, the PI3K, glycolysis, PPP, lipogenesis, glutaminolysis and pyrimidine synthesis pathways are notably altered in *human* HCC tumors (Fig. 9a). Importantly, ChREBP mRNA levels were significantly increased among HCC tumors harboring alterations in these pathways (Fig. 9a). This suggests that targeting ChREBP in these tumors might provide unique chemical probes and potential anti-cancer therapeutics. To test this hypothesis, ChREBP expression was stably inhibited in SNU475, Huh7 and BNL CL.2 cell lines (Supplementary Fig. 13a–l). For all these 3 cell lines, ChREBP silencing significantly reduced cell proliferation in vitro and suppressed xenograft tumor growth in vivo (Supplementary Fig. 13a–l). These results further highlight the potential therapeutic benefit of targeting ChREBP during cancer treatment when classical oncogenes or tumor suppressors are altered during HCC development. In this context and based on genomic sequencing data available from the TCGA - LIHC dataset, TERT, TP53, Axin1, ARID1A, cmyc and Jarid1B are the most frequently altered oncogenes or tumor suppressors in HCC (Fig. 9b). Therefore, to evaluate whether ChREBP was indeed required during HCC development, in cancer cells harboring alterations in these key identified drivers of *human* hepatocarcinogenesis, we stably overexpressed TERT, cmyc, jarid1b or stably inhibited p53, axin1 or Arid1a in the liver of either wild type (WT) or liver-specific ChREBP knockout *mice* (KO), using the sleeping beauty (SB) transposon system and the CRISPR/Cas9 approaches (Fig. 9c, d). As a result, overexpression of TERT, cmyc, jarid1b or knockdown of TP53, axin1 or Arid1a was sufficient to induce HCC development in WT *mice* (Fig. 9c, d). Strikingly, liver specific ChREBP deficiency was however able to significantly delay HCC burden, with only scattered liver preneoplastic lesions being detected in response to the activation these oncogenes or inhibiton of these tumor suppressors (Fig. 9c, d). Consequently, ChREBP knockout drastically improved the overall survival rate of *mice* overexpressing TERT, cmyc, JARID1B or of *mice* that were deficient for TP53, axin1 or arid1a (Supplementary Fig. 13m and Source data file of Supplementary Fig. 13). Finally, to confirm that ChREBP represents an interesting target to treat HCC, tumor development was induced by treating WT and liver-specific ChREBP KO *mice* with a combination of the carcinogen diethylnitrosamine (DEN) with the carbon-tetrachloride (CCl4) hepatotoxin. This chemical model of liver carcinogenesis has the particularity to mimic chronic liver inflammation and fibrosis that are observed during *human* HCC development[32]. In this inflammation-related liver-tumorigenic model, liver-specific ChREBP deficient *mice* also displayed a profound reduction in tumor development (number and size) (Fig. 9e). Consequently, ChREBP knockdown significantly increased the overall survival rate of *mice* in response to DEN/CCL4 treatment up to 320 days (Fig. 9f and Source data file of Fig. 9).

## The ChREBP pharmacological inhibitor SBI-993 effectively suppresses cell proliferation and tumor development

To further investigate the potential benefit of inhibiting ChREBP activity during HCC development, we took advantage of the small-molecule SBI-993, recently identified as a potent inhibitor of ChREBP activity[33]. We first showed that SBI-993 treatment inhibited ChREBP nuclear translocation, ChREBP activity and consequently the glucose-mediated induction of its target genes (Supplementary Fig. 14a–c). More importantly, SBI-993 significantly reduced SNU449 cell proliferation in a dose-dependent manner without inducing cell apoptosis when used up to 300 μM (Supplementary Fig. 14d, e). Supporting the hypothesis that pharmacological inhibition of ChREBP activity is of interest for cancer treatment, SBI-993 also significantly reduced SNU475, Huh7 and BNL CL.2 cell proliferation and xenograft tumor development when nude *mice* were treated daily with 50 mg/kg of SBI-993 for 3 consecutive weeks (Fig. 9g–i and Supplementary Fig. 14f, g). In the same line of evidence, daily injection of SBI-993 at 50 mg/kg for 3 weeks, inhibited ChREBP transcriptional activity in vivo when this transcription factor was overexpressed through adenoviral gene delivery in the liver of WT *mice* (Supplementary Fig. 14h). Consequently, SBI-993 abolished the induction of all identified pro-oncogenic ChREBP regulated genes, and therefore impaired ChREBP pro-proliferative effect (Supplementary Fig. 14i, j, k). Importantly, this 3-week challenge with SBI-993 at 50 mg/kg, that was administered daily for toxicology analysis, revealed apparent good tolerability, as illustrated by normal complete blood count and differential, normal blood chemistry and normal kidney or liver function (Supplementary Data 9). Even more striking, SBI-993, injected twice a week at 50 mg/kg up to 320 days, drastically reduced HCC development in response to TERT, c-myc or JARID1B overexpression, or in response to p53, axin1 or arid1a knockout (Fig. 9c, d). In the same line of evidence, SBI-993 treatment also delayed HCC development in response to DEN/CCL4 treatment (Fig. 9e). Overall, SBI-993 treatment improved mouse survival in all these HCC models by lowering HCC burden (Fig. 9f, supplementary Fig. 13m). After 320 days, no obvious toxic effect was observed in response to SBI-993 treatment (Supplementary Data 9). Collectively, these findings highlight the potential of ChREBP small-molecule inhibitors, specifically SBI-993, as anti-cancer therapeutics with acceptable tolerability.

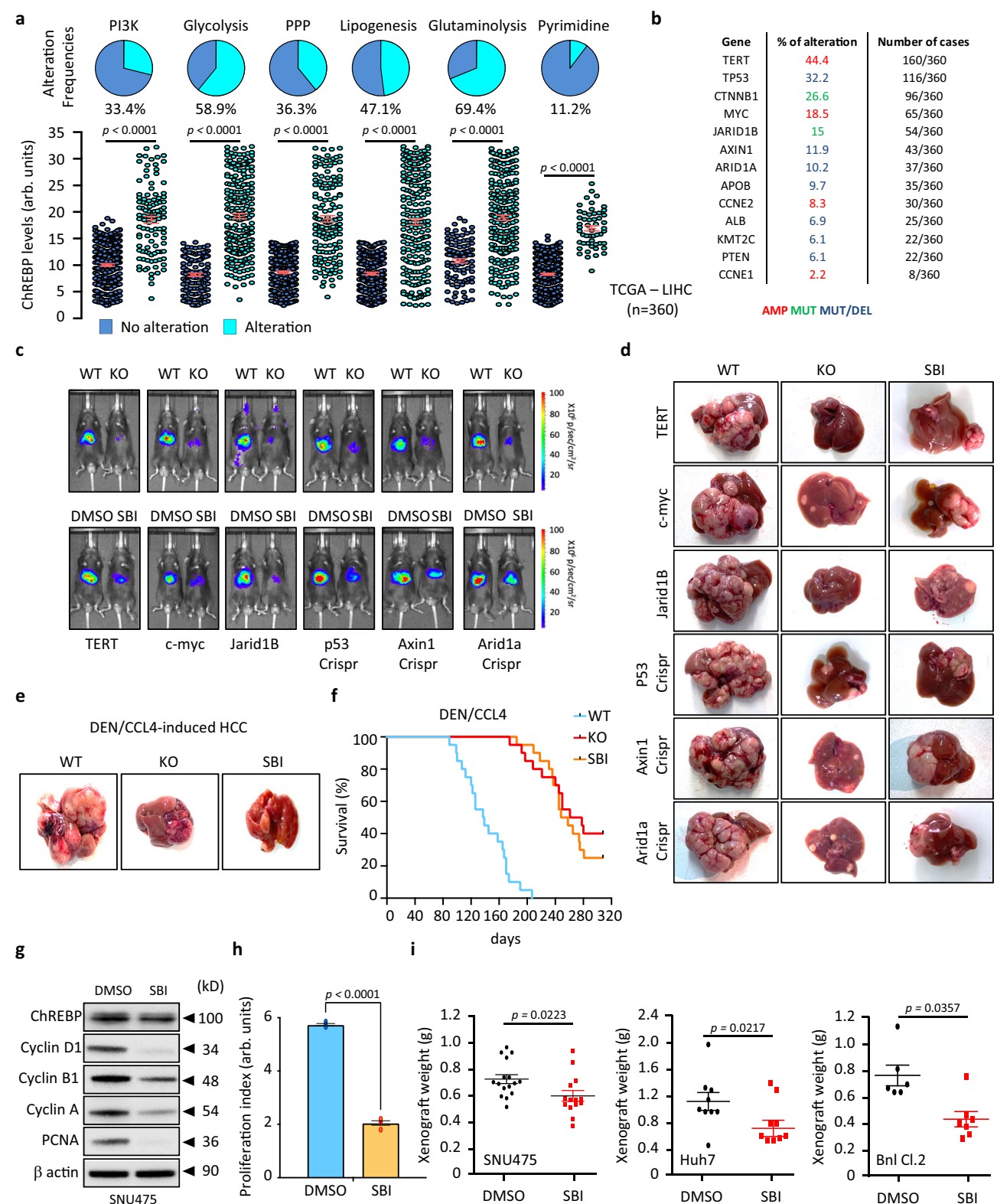

## ChREBP expression is inversely correlated with HCC sensitivity to sorafenib and its inhibition by SBI-993 reverses this drug resistance

Drug resistance is an additional important factor that determines poor prognosis of HCC patients. While sorafenib is one of the gold standard systemic drugs for treating patients with advanced stage HCC, its efficacy is severely affected by drug resistance[34]. This resistance has been particularly attributed to aberrant PI3K/AKT signaling and metabolic reprogramming[35]. Because ChREBP-mediated HCC development involved enhanced PI3K/AKT signaling in a p85α dependent-manner and profound metabolic rerouting, the correlation between ChREBP expression and the expression of its regulated genes with sorafenib resistance was analyzed from the publicly available dataset GSE109211[36]. The expression of ChREBP and its regulated genes was

**Fig. 9 | ChREBP represents a promising target for HCC treatment. a** In the TCGA-LIHC cohort, correlation of ChREBP expression between tumors with or without alterations in PI3K/AKT pathway, glycolysis, PPP, lipogenesis, glutaminolysis and pyrimidine synthesis. Upper panel displays alteration frequencies of the canonical pathways. Bottom panel demonstrates that HCC tissues carrying alteration of these pathways exhibited higher ChREBP expression levels. **b** Frequency of alteration of main oncogenes or tumor suppressors in HCC from the TCGA LIHC cohort. Amplification (AMP), mutation (MUT) and deletion (DEL) (**c, d**) HCC development was induced in WT or liver-specific ChREBP deficient *mice* (KO) by stably over-expressing either TERT, c-myc or jarid1B using the SB transposon system. HCC development was also induced by stably inhibiting the expression of p53, axin1 or arid1a using hydrodynamic injections. In these genetic models of HCC development, SBI-993 was also injected twice a week at 50 mg/kg in WT *mice* to determine whether pharmacological inhibition of ChREBP can reduce tumor development. **c** Representative bioluminescent imaging depicting tumor development in response to TERT, c-myc or jarid1B overexpression or in response to p53, axin1 or arid1a knockdown (*n* = 20 biologically independent *mice* per group). **d** Representative macroscopic images of HCC development in response to TERT, c-myc or jarid1B overexpression or in response to p53, axin1 or arid1a knockdown (*n* = 20 biologically independent *mice* per group). **e, f** HCC development was induced in either WT or liver specific ChREBP deficient *mice* by i.p injection of DEN (100 mg/kg) at day 15 postpartum followed by 25 biweekly i.p injections of CCl4 (0.5 ml/kg). *Mice* were studied at 12 months (*n* = 20 biologically independent *mice* per group). SBI-993 was also injected twice a week at 50 mg/kg in WT *mice* during DEN/CCL4 treatment to determine whether pharmacological inhibition of ChREBP can reduce tumor development (*n* = 20 biologically independent *mice* per group). **e** Representative images of HCC development shown (*n* = 20 biologically independent *mice* per group). **f** Kaplan−Meier analysis depicting *mice* overall survival rate (*n* = 20 biologically independent *mice* per group). Number of *mice* at risk at the specific time point can be found in the Source Data file. WT vs SBI *p* < 0.0001. WT vs KO *p* < 0.0001. KO vs SBI *p* = 0.635. **g** Representative Western blot analysis of cell cycle proteins in parental SNU475 cells treated with SBI-993 (20 μM) (*n* = 3 independent experiments). **h** Proliferation index of SNU475 cells (*n* = 3 independent experiments). **i** Xenograft mouse model using parental SNU475, Huh7 and Bnl Cl.2 cells. After engraftment, Nude *mice* were daily injected with SBI-993 (50 mg/kg) for 3 weeks. Tumor weight shown at 3 weeks (SNU475 *n* = 16, Huh7 *n* = 9, Bnl Cl.2 *n* = 6 biologically independent *mice* per group). All error bars represent mean ± SEM. **a**, **h**, **i** Statistical analyses were determined by unpaired two-sided Student's *t* test. **f** Significant difference in survival between cohorts was calculated using the log-rank (Mantel Cox) test. Source data are provided as a Source Data file.

significantly increased in non-responder (sorafenib-resistant) compared to responder (sorafenib-sensitive) patients (Fig. 10a). Furthermore, the expression of genes previously associated to sorafenib resistance in *human*, was also enhanced in ChREBP tumors (Fig. 10b). Accordingly, SNU475 and SNU449 cell lines stably overexpressing ChREBP were resistant to sorafenib-induced cell apoptosis (Fig. 10c), overall suggesting that upregulation of ChREBP activity may account for sorafenib resistance. To test this hypothesis, an in vitro model that recapitulates sorafenib resistance was used to investigate the role of ChREBP in this process. Thus, 8 months treatment of parental SNU449, SNU475, Huh7 and SK-Hep1 HCC cells with increased concentration of sorafenib generated sorafenib resistant (SR) clones that were resistant against sorafenib-mediated cell-induced apoptosis when sorafenib was used at 15 μM (Fig. 9d and Supplementary Fig. 15a). Interestingly, resistance against sorafenib treatment was correlated with increased ChREBP expression, enhanced PI3K/AKT signaling and metabolic reprogramming in all resistant cell lines (Supplementary Fig. 15b, c, d). As previously observed in this study, increased ChREBP activity was overall associated with enhanced expression of genes involved in glycolysis, PPP, de novo lipogenesis, glutaminolysis and pyrimidine biosynthesis (Supplementary Fig. 15c, d). Mechanistically, studies have previously described that sorafenib treatment significantly impaired mitochondrial respiration[37,38]. As a result, the subsequent activation of the PI3K/AKT signaling pathway boosted aerobic glycolysis (Warburg shift) in sorafenib-resistant cancer cells with defective mitochondria oxidative capacity to protect cell from sorafenib-induced cell apoptosis. Therefore, we investigated whether such a Warburg shift, a hallmark of cellular transformation, could be the underlying mechanism of resistance in our model and to which extend ChREBP was involved in this process. The oxygen consumption rate (OCR), which reflects mitochondrial oxidative phosphorylation (OXPHOS) capacity, was first measured in SR-SNU449 and SR-SNU475 cells 6 hours after sorafenib (15 μM) treatment. As shown in Supplementary Fig. 16a, b, sorafenib led to a severe reduction of the basal OCR, ATP-linked OCR, and maximal respiration rate compared to untreated SR-SNU449 and SR-SNU475 cells. Because the effect of sorafenib on mitochondrial ATP-linked OXPHOS was drastic, we reasoned that the cells might execute a shift toward aerobic glycolysis to enable survival. Thus, we also monitored glycolytic function upon acute sorafenib treatment. Sorafenib treatment of SR-SNU449 and SR-SNU475 cells significantly increased ECAR, a proxy for glycolytic activity, compared to untreated SR clones (Supplementary Fig. 16c, d). This glycolytic alteration thus sustained ATP production in SR clones despite the important reduction in mitochondrial ATP production rate (Fig. 10e and

Supplementary Fig. 16e). In sorafenib-treated SR clones, this Warburg shift was associated with resistance to apoptosis (Fig. 10f, g and Supplementary Figs. 15e, f, g). However, cells treated with SBI-993 demonstrated no metabolic flexibility when exposed to sorafenib. In fact, co-treatment with sorafenib and SBI-993 reduced both mitochondrial oxidative capacity and glycolytic function of SR clones, compromising SNU449 and SNU475 ATP production (Fig. 10e and Supplementary Fig. 16e). Overall, these data demonstrated that, while sorafenib induced persistent OXPHOS inhibition eliciting a Warburg shift to glycolysis, pharmacological inhibition of ChREBP by SBI-993 can acutely curtail this shift, thereby reducing cancer cell survival by restoring sensitivity to sorafenib treatment (Supplementary Fig. 16f). Furthermore, SBI-993 inhibited the increase in the PI3K/AKT signaling pathway observed in sorafenib resistant HCC cells, which will help to restore sensitivity to sorafenib treatment (Supplementary Fig. 15h). Overall, our data indicate that sorafenib acts noncanonically as an OXPHOS inhibitor, initiating a Warburg shift to aerobic glycolysis in resistant HCC cancer cells. However, ChREBP pharmacological inhibition restrains this metabolic flexibility by inhibiting glucose metabolic rerouting, PI3K/AKT signaling and, with that, confers sensitivity toward the anticancer effect of sorafenib (Supplementary Fig. 16f). To further test the potential interest of inhibiting ChREBP activity using SBI-993 to improve sorafenib efficiency, we xenografted parental and SR-resistant SNU449, SNU475, Huh7 and SK-Hep1 HCC cells in nude *mice* and subsequently subjected them to sorafenib and/or SBI-993 treatment. Sorafenib (10 mg/kg) was given orally every 2 days using Captisol (30% in water) as vehicle. SBI-993 (50 mg/kg) was injected alone or in combination with sorafenib. Strikingly for all cell lines tested, SBI-993 was able to totally restore sorafenib sensitivity against these sorafenib resistant clones (Supplementary Fig. 17a). After engraftment and waiting 21 days for the tumors to grow, this co-treatment ultimately resulted in a drastic reduction of xenograft tumor volume in all *mice*. Interestingly, SBI-993 alone was also a potent inhibitor of cell proliferation, which significantly delayed tumor growth for all sorafenib resistant clones. Based on these results, we finally confirmed the synergistic effect of sorafenib and SBI-993 treatment using our DEN/CCL4 induced-HCC model as previously described. Sorafenib and SBI-993 administration was initiated at 6 months, a time point where HCC formation was established in all *mice*. Again sorafenib (10 mg/kg) was given orally every 2 days using Captisol (30% in water) as vehicle. SBI-993 (50 mg/kg) was injected alone or in combination with sorafenib. The effect on tumor development was assessed 3 months after the beginning of this combinational treatment. Sorafenib or SBI-993 alone attenuated liver tumor growth as evidence by the significant reduction

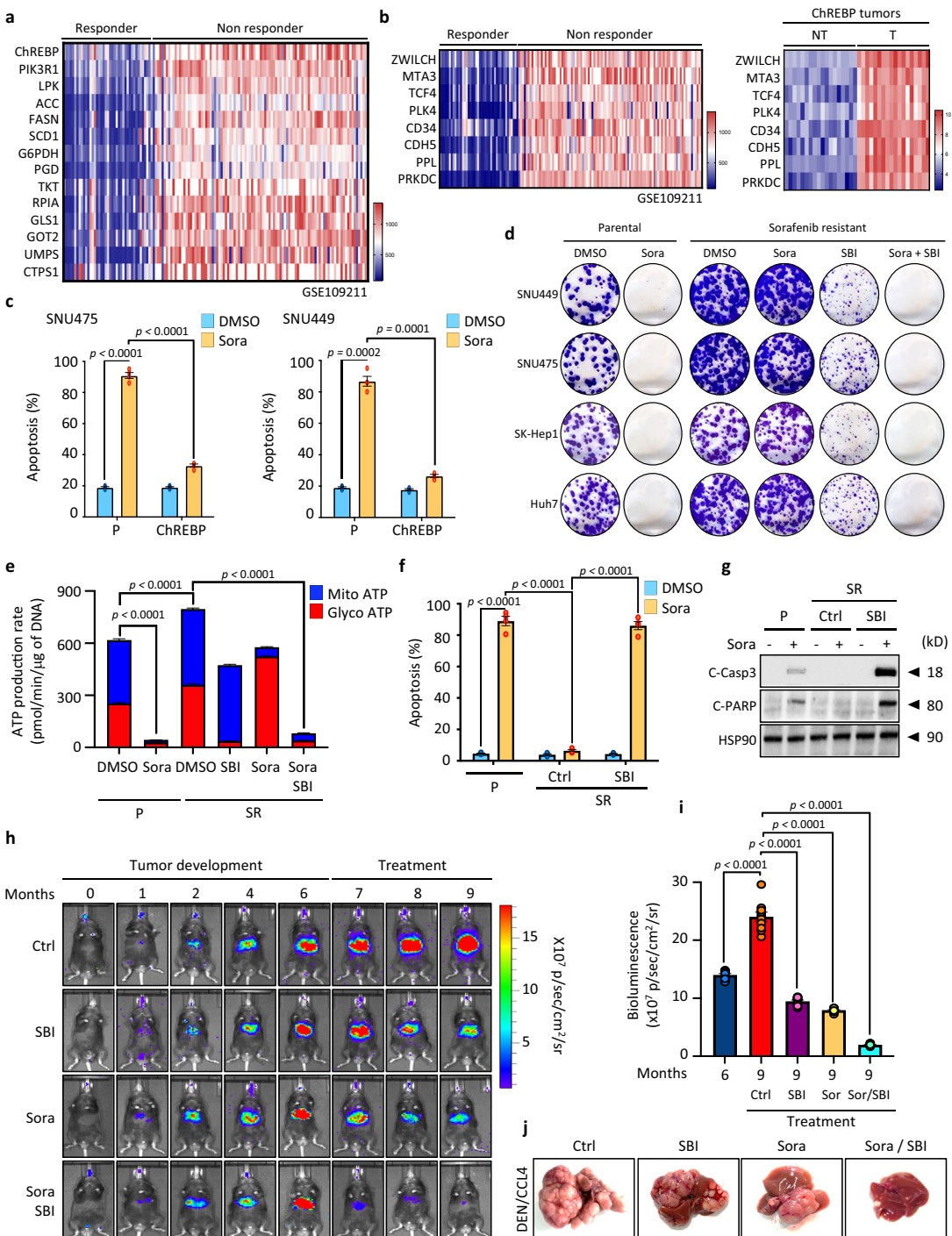

**Fig. 10 | Enhanced ChREBP activity participates to sorafenib resistance. a** Data mining of the expression levels of ChREBP and ChREBP-regulated genes in the form of an heatmap, between sorafenib responder and non-responder patients from the dataset GSE109211. **b** Expression of genes associated with sorafenib resistance, in the form of an heatmap, in responder and non-responder patients from the dataset GSE109211 and in ChREBP tumors (T) compared to non-tumoral tissues (NT) ($n = 18$ biologically independent *mice* per group). **c** Caspase 3/7 activity measured in parental or in ChREBP overexpressing SNU475 (left panel) and SNU449 (right panel) in response to 15 μM sorafenib treatment for 24 h ($n = 4$ independent experiments). **d** Representative clonogenic assays showing the effect of sorafenib (15 μM) with or without SBI-993 (40 μM) treatment on sorafenib resistant SNU449, SNU475, SK-Hep1 and Huh7 cells ($n = 3$ independent experiments). **e** ATP production rate from glycolysis or oxidative phosphorylation in parental or sorafenib resistant SNU449 cells ($n = 9$). P (parental), SR (sorafenib resistant clone), Sora (Sorafenib). **f** Effect of SBI-993 and sorafenib co-treatment on SR-SNU449 cell apoptosis ($n = 4$

independent experiments). **g** Representative Western blot depicting caspase 3 and PARP activity in SR-SNU449 cells treated with sorafenib alone or in combination with SBI-993 ($n = 3$ independent experiments). **h–j** HCC was induced in C57bl6/J *mice* by i.p injection of DEN (100 mg/kg) at day 15 postpartum followed by 24 biweekly i.p injections of CCl4 (0.5 ml/kg). At 6 months of age, *mice* were then treated with sorafenib alone or in combination with SBI-993 for 3 months. Sorafenib (10 mg/kg) was given orally using Captisol (30% in water) as vehicle. SBI-993 was i.p injected at 50 mg/kg. **h** Representative bioluminescence imaging depicting tumor development in response to sorafenib and SBI-993 treatment ($n = 10$ biologically independent *mice* per group). **i** Quantification of bioluminescence activity reflecting tumor development shown ($n = 10$ biologically independent *mice* per group). **j** Representative macroscopic images of HCC development after sorafenib and or SBI-993 treatment ($n = 10$ biologically independent *mice* per group). All error bars represent mean ± SEM. Statistical analyses were made using two-way ANOVA and Tukey's multiple-comparisons test. Source data are provided as a Source Data file.

in bioluminescence activity over time and smaller tumor diameter at the end of the treatment compared to control *mice* (Fig. 10h–j). More importantly, combined treatment with sorafenib and SBI-993 exerted a more potent anti-tumor effect against HCC compared to treatment with sorafenib or SBI-993 alone (Fig. 10h–j). These results thus indicate that SBI-993 enhances sorafenib-induced cell growth inhibition and induction of cell apoptosis to restrain tumor development. In conclusion, our results identify ChREBP as a valuable indicator of sorafenib responsiveness and a useful marker for the clinical identification of patients who would benefit from sorafenib treatment. Thus, pharmacological inhibition of ChREBP using SBI-993 constitutes a valuable therapeutic opportunity for HCC treatment (Supplementary Fig. 17c).

## Discussion

Targeting specific metabolic pathways whose activity is required for tumor growth is an emerging therapeutic approach in experimental and clinical oncology. Most cancer cells must cope with specific metabolic adaptations to promote their survival to support their unrelenting cell division associated with uncontrolled oncogenic signaling pathways. According to new evidence, abnormal glycolysis, de novo lipogenesis, pentose phosphate shunt, glutaminolysis, and de novo pyrimidine biosynthesis all play important roles in malignant transformation[3,4,39]. Thus, identifying genes whose properties promote cancer growth by regulating cell metabolic fitness is critical for the development of novel, specific, and effective antineoplastic therapies. ChREBP appears to be a potential driver of cancer development in this scenario because it is a central regulator of metabolism in various cell types in response to environmental and hormonal stimuli[40]. However, these studies have underscored the multifaceted role of ChREBP in controlling HCC initiation and development. The current study provides the first evidence that ChREBP is a new oncogene in liver cancers. The observation that ChREBP activity is increased in HCC from multiple patient datasets, especially in tumors with an unfavorable outcome, adds to the *human* relevance of ChREBP oncogenic function. Importantly, our findings show that ChREBP is a new molecular biomarker that predicts resistance to systemic therapy during HCC treatment.

At the molecular level, we provide evidence that ChREBP is required for appropriate reprogramming of glucose and glutamine metabolism to support HCC tumor growth. We report that ChREBP coordinates glucose sensing with glutamine utilization and subsequent downstream biosynthetic pathways to sustain tumor growth. ChREBP downstream metabolic effectors identified in this study first include those involved in de novo lipid biosynthesis and fatty acid desaturation. Interestingly, since the inhibition of enzymes involved in these metabolic processes attenuates cancer cell growth, targeting lipid metabolism is increasingly attracting for therapeutic strategy[41]. Our findings support this theory because ChREBP overexpressing tumors reroute glucose carbons toward lipid synthesis with a specific enrichment of MUFA in all classes of phospholipids. This increase in MUFA may confer several advantages to highly proliferative cancer cells where ChREBP activity is enhanced[42]. One such advantage would be to prevent saturated fatty acid accumulation from increased lipogenesis, which could trigger otherwise apoptotic signaling pathways[41]. Furthermore, MUFA-PLs have been also reported to increase membrane fluidity in several cancer types, resulting in enhanced activation of specific oncogenic signaling pathways to sustain tumor growth[43]. For example, MUFA-PLs have been demonstrated to enhance IGF1R and IR activity in several type of cancers. Our study thus reinforces the concept that changes in the abundance of multiple lipid metabolic genes, in response to ChREBP activation, are associated with HCC development and progression.

In addition to the regulation of fatty acid synthesis, our study also demonstrates that ChREBP coordinates glucose and glutamine utilization to support de novo nucleotides synthesis. In this process, ChREBP activation, by directly regulating the expression of key enzymes of the PPP, first reroutes glucose metabolic fluxes to sustain de novo nucleotides synthesis. These observations are particularly important since alterations in the PPP play an important role in metabolic reprogramming and HCC tumor development[44]. This pro-proliferative effect is mainly mediated through the production of ribose 5-phosphate (R5P) and its subsequent transformation into PRPP, which in turn is used as a precursor for nucleotide production[45]. In this context, while ChREBP's connection with enhanced pentose phosphate shunt was previously suggested, one important observation of our work was to identify ChREBP as a major player of glutaminolysis and de novo pyrimidine biosynthesis[46]. Our study shows that ChREBP-driven HCC cancer cells also depend on glutamine metabolism for proliferation. ChREBP activation, through the direct transcriptional regulation of GLS1, GOT2, CTPS1 and UMPS, facilitates the conversion of glutamine to glutamate, leading preferentially to cytosolic conversion of aspartate into N-carbamoylaspartate to serve as substrate, with PRPP, for pyrimidine synthesis. In a context of therapeutic treatment, individual inhibition of these newly identified ChREBP target genes was shown to impact pyrimidine synthesis and cancer development[47]. In this context, our data highlight that ChREBP inhibition is an interesting approach to inhibit nucleotide synthesis in proliferating cancer cells. Interestingly, the pyrimidine pool imbalance over purine synthesis (increase in the ratio of pyrimidine to purine metabolites), observed in response to ChREBP activation, is particularly of interest since it contributes to genetic instability in cancer[48]. This imbalance, known as a "pyrimidine-rich transversion mutational bias," promotes mutagenesis and cell transformation by increasing the bias toward R/Y (purine to pyrimidine) versus Y/R point mutations on the DNA sense strand (PTMB)[49]. According to our findings, the diversion of glutamine carbons into aspartate production favors the first reactions in the pyrimidine synthesis pathway and causes an imbalance in the pyrimidine pool. Collectively, our data describe a unique metabolic circuitry under the control of ChREBP that maintains nucleotide biosynthesis to support HCC development. Overall, ChREBP by increasing the conversion of aspartate into pyrimidine may contribute to increase DNA genomic instability and mutagenesis to promote cell transformation.

This specific reprogramming of glucose and glutamine metabolism under the control of ChREBP seems to involve a shared and limited set of pathways that commonly support tumor growth. Interestingly, it has been proposed that these metabolic modifications not only support cancer cell survival and proliferation but also interact with oncogenic signaling pathways such as the PI3K/AKT signaling, to enhance the invasive and metastatic properties of cancer cells[50]. In this line of evidence, pathway screening also identifies the PI3K/AKT cascade as the major downstream signaling mechanism underlying the oncogenic effect of ChREBP in HCC initiation and development. Ectopic expression of ChREBP markedly activated the PI3K/AKT pathway in a mechanism that relies on the activation of the IGF1R and IR activity and the transcriptional control of PIK3R1, which encodes the regulatory subunit p85α of the class 1A PI3K[51]. Recent studies have suggested that p85α acts as an oncogene, since mutations in the inter-Src homology-2 (iSH2) domain of p85α isoform are frequently detected in colon and primary endometrial cancers[52–54]. Furthermore, the expression of PIK3R1 is also significantly increased during HCC development, thus facilitating cell proliferation and migration in a PI3K/AKT dependent manner[55,56]. Consequently, controlling p85α expression may overcome such activation of the PI3K activity during cancer development. Supporting this hypothesis, inhibiting PIK3R1 expression in ChREBP tumors abolished ChREBP-mediated increase in PI3K/AKT signaling and cell proliferation. Therefore, inhibition of ChREBP transcriptional activity might be very effective in combinatorial treatments with PI3K inhibitors to achieve synergistic effect in treating such PI3K/AKT activated tumors. Accordingly, our study

demonstrated that ChREBP overexpression alone was not sufficient to trigger its activation in the absence of proper PI3K/AKT signaling. This important finding thus led us to question the existence of a positive amplification feedback loop in which ChREBP-mediated p85α expression and subsequent activation of the PI3K/AKT signaling pathway strengthen ChREBP activation itself by promoting its nuclear translocation. We demonstrated for the first time that HK2 is a component of an amplification loop that links glucose-mediated ChREBP activation with increased PI3K/AKT signaling. We have clearly found that ChREBP activation, by increasing the PI3K/AKT pathway, stimulates HK2 expression and activity, which in turn, by phosphorylating glucose into G6P, contributes to ChREBP nuclear translocation and activation itself. The importance of HK2 during HCC development has been previously reported since liver specific HK2 deficiency decreases HCC cell proliferation and tumor development in response to DEN treatment. In *mice*, HK2 depletion inhibits glycolysis and induces OXPHOS, enhancing the sensitivity of HCC to systemic drugs, such as sorafenib, thereby inhibiting tumor growth[57]. Collectively, our results further reinforce the concept that HK2 can connect and amplify growth factors signaling with ChREBP-mediating regulation of glucose metabolism to support cell proliferation and cancer development[46]. Overall, ChREBP appears to be at the crossroad of many important biological pathways and processes involved in neoplastic cell growth and proliferation. Our study particularly demonstrates that ChREBP serves as a molecular link between growth signals (IGF1R, IR activity and PI3K/AKT signaling) and acute control over determinant intertwined metabolic pathways.

Since ChREBP activation is an important feature of hepatic cancer cell proliferation and survival provides a unique opportunity for the development of new therapeutics for HCC treatment. Since the main mechanism regulating ChREBP activity involved its shuttling between the cytosol and nucleus and its conformational change by metabolites, several reports have suggested that its nuclear localization could be inhibited by "starvation signals". Various metabolites including ketone bodies, 3-hydroxybutyrate, and acetoacetate, which are elevated during fatty acid mobilization, have been identified to favor the sequestration of ChREBP within the cytosol[58–60]. In addition, an increase in AMP levels, which occurs in liver during fasting in conjunction with a decrease in the ATP/ADP ratio, also favors the sequestration of ChREBP in the cytoplasm[61]. Collectively, these results suggest that AMP and ketone bodies might represent two regulatory mechanisms contributing to cytosolic ChREBP sequestration during periods of fasting or ketosis. Even though direct evidence is still needed to support this hypothesis in a context of HCC treatment, these mechanisms of regulation during low carbohydrate or ketogenic diets may result in the reduction of HCC development by inhibiting ChREBP activity. Nevertheless, pharmacological strategies have also emerged to inhibit ChREBP function. One strategy directed toward interrupting ChREBP-importin α complex stabilization, and consequently blocking ChREBP function, has been developed by the group of Professor Uyeda[62]. Similarly, a potent stabilizer of the interaction between the 14-3-3 protein and ChREBP also proved its efficiency to inhibit ChREBP transcriptional activity[63]. These studies support the feasibility of developing effective and specific inhibitors of ChREBP. However, evidence for in vivo efficacy was lacking in these two studies. In this context, our study is the first to show that inhibiting ChREBP transcriptional activity in HCC is effective in reducing HCC tumor growth. We indeed unravel that SBI-993, displaying a potent inhibitory effect on ChREBP activity, represents a promising chemical molecule with significant in vivo anti-tumor potential with good tolerability. We found that SBI-993 exerts its anti-proliferative effect by markedly reducing ChREBP nuclear translocation and its transcriptional activity. The ChREBP-mediated mechanism described in our study is based on the work led by Doctor Daniel P. Kelly's group using a cell-based phenotypic small-molecule screen[33]. In this study, SBI-993, by controlling the activity of the transcription factor *MondoA* (*mlxip*),

which belongs with ChREBP (*mlxipl*) to the Mondo family of transcription factors, was identified as an inhibitor of de novo lipid synthesis in skeletal muscle. Analysis of MondoA and ChREBP respective tissue distribution reported widespread expression patterns for these paralogs. ChREBP predominates in liver, as well as in white and brown adipose tissues, whereas MondoA expression is most abundant in skeletal muscle, suggesting distinct functions. Accordingly, while MondoA limits glucose uptake and glycolysis, ChREBP promotes glycolysis and de novo lipogenesis. Moreover, clinical investigations have shed light on MondoA and ChREBP expression profiles during *human* malignancies. Increased occurrence of MondoA deficiency in several tumors, including lymphomas, sarcomas, bladder, and colon carcinomas[24,64–67], in parallel to the suppression of MondoA recruitment at the Txnip or Arrdc4 promoter upon various oncogenic drivers (Ras, BRAF, PI3K and mTOR) suggests that MondoA silencing may contribute to the transformed phenotype across broad cancer types[64,68,69]. By contrast, increased ChREBP expression levels in response to mitogenic stimulation, as well as in various tumors, including prostate and breast cancers, suggest that ChREBP signaling may be activated during carcinogenesis[46,70,71]. Based on these observations, it is tempting to speculate that SBI-993 anti-proliferative effect is likely the result of ChREBP, rather than MondoA, transcriptional inhibition. Supporting this conclusion, we found that MondoA expression was not affected in HCC from multiple patient datasets. Overall, whereas this approach was powerful in what we believe to be a novel regulatory mechanism relevant to the control of HCC development, SBI-993 specificity of action remains to be fully determined in the future. It is possible that, in addition to the modulation of ChREBP activity, SBI-993 can directly target enzymes involved in HCC metabolic reprogramming or in modulation of the PI3K/AKT signaling. However, our results are most consistent with a mechanism that imposes upstream control of ChREBP signaling as SBI-993 abolished ChREBP effect on metabolic reprogramming and modulation of the PI3K/AKT signaling. Our results thus demonstrate that SBI-993 may affect the level of an intracellular signal that controls ChREBP nuclear localization, but future studies are necessary to tackle SBI-993 specificity in regard with cancer treatment. Nevertheless, our findings unambiguously support the clinical development of ChREBP inhibitors as a therapy for the treatment of *human* cancers. Supporting this concept, SBI-993 also significantly reduced the proliferation of colorectal and breast cancer cells, as well as leukemic T cells (Supplementary Fig. 17b). Such inhibitors could also represent important pharmacological tools to overcome resistance against systemic therapies as we demonstrate that ChREBP represents in HCC a new molecular biomarker that predicts sorafenib responsiveness. In this context, SBI-993, by preventing the activation of oncogenic signaling pathways and compensatory cell metabolic rewiring, is likely to contribute to sensitize cancer cells to HCC systemic treatments. In conclusion, our study proposes that targeting ChREBP activity may serve as an anti-neoplastic strategy and as a promising approach to counteract resistance against systemic therapy during HCC treatment (Supplementary Fig. 17c).

## Methods

Our research complies with all relevant ethical regulations and institution that approved the study protocol.

### Antibodies

The following antibodies were used in this study for either western blotting, ChIP-sequencing, immunohistochemistry and chromatin immunoprecipitation. ChREBP (Novus, NB400-135, 1/1000[e]), SCD1 (Cell Signaling, C12H5, 1/1000[e]), LPK (Abcam, ab6191, 1/1000[e]), ACC (Cell Signaling, 3662, 1/1000[e]), FAS (Cell signaling, C20G5, 1/1000[e]), G6PDH (Cell Signaling, 12263, 1/1000[e]), GAPDH (Santa Cruz Biotechnology, FL-335, 1/1000[e]), PKM2 (Cell Signaling, 4053, 1/1000[e]), GOT1 (Cell Signaling, 344423, 1/1000[e]), G6PDH (Cell Signaling,

8866, 1/1000[e]), GOT2 (Novus Biologicals, NBP1-47469, 1/500[e]), HK2 (Cell Signaling, 2106, 1/1000[e]), PGD (Novus Biologicals, NBP1-31589, 1/1000[e]), RPIA (Novus Biologicals, NBP1-86214, 1/1000[e]), ATP-Citrate Lyase (Cell Signaling, 4332, 1/1000[e]), SCD1 (Cell Signaling, 2794, 1/1000[e]), SLC1A5 (Novus Biologicals, NBP1-89327, 1/1000[e]), Gls1 (Novus Biologicals, NBP1-89766, 1/1000[e]), Gls2 (Novus Biologicals, NBP1-76544, 1/1000[e]), GPT (Novus Biologicals, NBP1-89111, 1/500[e]), CAD (Cell Signaling, 11933, 1/1000[e]), P-CAD Pser1859 (Cell Signaling, 12662, 1/1000[e]), UMPS (Novus Biologicals, NBP1-85896, 1/1000[e]), CTPS1 (Novus Biologicals, NBP1-52892, 1/1000[e]), PRPS2 (Novus Biologicals, NBP1-31435, 1/1000[e]), p(S473)-AKT (Cell signaling, D9E, 1/1000[e]), Akt (Cell Signaling, 9272, 1/1000[e]), P-p70S6K (Cell Signaling, 108D2, 1/1000[e]), p70S6K (Cell Signaling, 49D7, 1/1000[e]), p-GSK3β (Cell Signaling, D17D2, 1/1000[e]), GSK3β (Cell Signaling, 27C10, 1/1000[e]), PTEN (Cell Signaling, 9552, 1/1000[e]), PI3 Kinase p85α (Cell Signaling, 13666, 1/1000[e]), Afp (Cell Signaling, 4448, 1/1000[e]), KI67 (Cell Signaling, 9449, 1/1000[e]), Cyclin D1 (Cell Signaling, 2922, 1/1000[e]). Cyclin B1 (Cell Signaling, 4138, 1/1000[e]), Cyclin E (Cell Signaling, 4129, 1/1000[e]), Cyclin A (Cell Signaling, 4656, 1/1000[e]), PCNA (Cell Signaling, 2586, 1/1000[e]), RNA PolII (Santa Cruz, sc899,), HSP90 (Cell Signaling, 4874, 1/1000[e]), H3K4me3 (Actif motif, 39915).

### Collection of *human* liver cancer microarray datasets

For this study, 10 published datasets of *human* liver HCC were used. Eight HCC microarray datasets publicly available (Supplementary Table 1) were assembled with ArrayExpress and the Gene Expression Omnibus (GEO) of the National Center for Biotechnology Information (NCBI)[16–23]. mRNA expression and clinical data from 334 HCC tumors from the LIHC dataset and 50 normal control samples were also obtained from The Cancer Genome Atlas (TCGA) (http://cancergenome.nih.gov/) and cBioportal for Cancer Genomics (http://www.cbioportal.org/). Main clinical data of this dataset are summarized in Supplementary Tables 2 and 3. Finally, a series of 160 HCC samples and their non-tumor counterparts (LICA-FR dataset), which were collected from patients surgically treated in four French hospitals located in Bordeaux and Paris region, were also obtained from the TCGA and cBioportal. HCC developed mostly on a non-cirrhotic liver (107/160, 67%): 75 tumors developed in non-fibrotic (METAVIR F0-F1), 32 in chronic hepatitis (F2−F3) and 53 in cirrhotic liver (F4). Clinicopathological data were available for all cases (Tables S2 and S4). These 160 samples were analyzed by RNA sequencing and the sequencing data have been published and deposited to the EGA (European Genome-phenome Archive) database (RNA-seq accession [EGAS00001002879][72]. Overall, 1058 *human* HCC gene expression profiles were compared with 630 control liver samples (Supplementary Table 1). Data were downloaded from the publicly available database hence it was not applicable for additional ethical approval.

### *Human* tissue samples

A collection of 12 frozen HCC and corresponding non-tumorous surrounding livers, from first line surgically treated patients, were used to study at the protein level the expression of ChREBP and MondoA in HCC tissues. The clinicopathological features of liver cancer patients are summarized in Supplementary Table 3. HCC specimens were collected at the Cochin Institute hospital. Institutional Review Board approval was obtained at the local Ethical Committee (CPP: 2015-08-11 MS3 DC). Informed consent was obtained from the participants both for the use of their samples and to publish information that identified individuals.

### Processing and identification of consistently altered genes

To eliminate analytical bias that might arise from data processing, the NCBI GEO2R tool was used to directly determine the differentially expressed genes between healthy or adjacent liver tissue samples with HCC samples in each dataset. Thereafter, the GEO2R outputs were downloaded, and all genes differentially regulated with the cut-off criteria of $P < 0.001$ and log2FC (fold-change) > 1.5 were selected. Then, to identified common altered metabolic gene signature, a previously published list of 2752 metabolism-annotated genes was used to extract only the deregulated metabolic genes in each of the 10 datasets (Supplementary Data 4)[73]. Furthermore, the average log of fold change (logFC) of all differentially expressed genes as determined by GEO2R was calculated and used as reference to set cutoff threshold values for each dataset. This step ensured the exclusion of metabolic gene probes with very small expression changes. For onward analyses, metabolic genes with þlogFC at or above the cutoff value in the respective datasets were selected as up-regulated, whereas those with −logFC at or below cutoff value were selected as down-regulated. Following these prior steps, a metabolic gene was identified as consistently altered if it has the same expression pattern (i.e., exclusively in the up-regulated or down-regulated category) in at least 8 of the 10 HCC datasets.

### Gene set enrichment analysis (GSEA)

Based on the entire microarray profiles using predefined gene sets, GSEA applies computational methods to detect pathways that may serve as targets for novel therapeutics[74]. The Hallmark gene set file was obtained from the MSigDB database. The gene sets included in the analysis were limited to those that contained between 10 and 500 genes. Permutation was conducted 1000 times according to default-weighted enrichment statistics and by using a signal-to-noise metric to rank genes according to their differential expression levels. Significant gene sets were defined as those with a nominal $P < 0.05$.

### Kaplan-Meier overall survival analyses

The LIHC and LICA-FR dataset, which were the largest cohort analyzed in this study and which contain recorded clinical data (Supplementary Tables 2, 3 and 4), were used for the overall survival analyses. Patient survival was followed for 5 years after HCC discovery. Prior to the study, the expression pattern of ChREBP was confirmed to be the same in LIHC and LICA-FR dataset than described for the 8 other microarrays. To ensure that the analyzed overall survival can be attributed to a difference in the expression of ChREBP, its expression values were used to rank the patients into lower, intermediate, and higher groups. Subsequently, patients with lower and higher expression values for ChREBP were adopted as a uniform inclusion criterion for survival prediction. Based on this criterion, overall survival (OS) was assessed, and Kaplan-Meier overall survival analysis was performed with log-rank (Mantel-Cox) test in GraphPad Prism. A statistical significance was accepted at $P < 0.05$.

### In vivo experiments using *mice*

All procedures using *mice* were carried out according to the French guidelines for the care and use of experimental animals (animal authorization agreement number CEEA34.RD.082.12, Paris Descartes Ethical Committee). *Mice* were maintained in a 12 h light/dark cycle with water and a standard diet SAFE® D113 (64.1% carbohydrate, 13.4% fat, and 22.5% protein). Only male *mice* were considered in this study because adenoviral infection and transfection of plasmid using the sleeping beauty system were performed through penis vein injection.

### Vector design for sleeping beauty (SB) transposon system

To generate the pT3-EF1a-ChREBP-IRES-Luciferase vector (ChREBP-luc), the pT3-EF1A-MYC-IRES-luc was used as a donor vector. The cloning of ChREBP, with N-terminal HA tag, into the donor vector was performed by In-Fusion cloning (Takara Bio, 638910). The pT3-EF1A-MYC-IRES-luc plasmid was a gift from Amaia Lujambio (Addgene plasmid # 129775). The CMV-SB13 transposase plasmid was kindly provided by Dr. Scott Lowe (MSKCC, New York). TERT, c-myc and Jarid1B were also stably overexpressed in the liver of C57BL6/J *mice* using the SB transposon system. These oncogenes were overexpressed

using the respective plasmids: pT3-EF1A-mTert (gift from Amaia Lujambio (Addgene plasmid # 162555)), PT3-EF1a-c-myc (gift from Xin Chen (Addgene plasmid # 92046)). To generate the pT3-EF1a-Jarid1b vector, the pT3TS vector was used as a donor vector (gift from Stephen Ekker (Addgene plasmid # 31830)). For these 3 oncogenes, the SB13 transposase was stably overexpressed using the PT2/C-Luc//PGK-SB13 plasmid which was a gift from John Ohlfest (Addgene plasmid # 20207). All constructs were verified by nucleotide sequencing and vector integrity was confirmed by restriction enzyme digestion. Vectors for hydrodynamic delivery were produced using the QIAGEN plasmid PlusMega kit (QIAGEN, 12981). Equivalent DNA concentration between different batches of DNA was confirmed to ensure reproducibility among experiments.

### Vector design for inhibiting in vivo the expression of specific gene with sgRNA using CRISPRi technology

A sgRNA expression plasmid was constructed for each targeted gene (PIK3R1, p53, Axin1, Arid1A) by ligating the corresponding annealed guide oligonucleotides to the pX330-U6-Chimeric_BB-CBh-hSpCas9 plasmid that co-expressed the Cas9[75]. The pX330-U6-Chimeric_BB-CBh-hSpCas9 was a gift from Feng Zhang (Addgene # 42230). pX330-U6-Chimeric_BB-CBh-hSpCas9 vector was digested with BbsI and ligated with annealed gRNA oligonucleotides. An extra G is added for sgRNAs lacking a 5′ G for U6 transcriptional initiation (mouse ARID1A gRNA: 5′GCGGTACCCGATGACCATGC3′, mouse PIK3R1 gRNA: 5′GCCC TGGGCTACTTACACTA3′). px330-Axin1 was a gift from Amaia Lujambio (Addgene plasmid # 162543). pX330-p53 was a gift from Tyler Jacks (Addgene plasmid # 59910). To ensure appropriate disruption of gene function, the Cas9 was also stably overexpressed with the sleeping beauty transposon system using the pT3TS-nCas9n vector which was a gift from Wenbiao Chen (Addgene plasmid # 46757). Vectors for hydrodynamic delivery were produced using the QIAGEN plasmid PlusMega kit (QIAGEN, 12981).

### Hydrodynamic injection

Hydrodynamic injection procedures were conducted as previously described[76]. FVB/N *mice*, obtained from Charles River, were used to determine the pro-oncogenic function of ChREBP in the liver. Liver-specific ChREBP deficient *mice*, previously characterized in[77], were used to determine the contribution of ChREBP during HCC development in response to TERT, c-myc, Jarid1B overexpression or in response to p53, axin1 and Arid1A knockdown. For hydrodynamic injection, 10 µg of pT3-EF1a-ChREBP-IRES-Luciferase vector (ChREBP-luc), pT3-EF1A-mTert, pT3-EF1A-myc and pT3-EF1a-Jarid1b constructs, in a ratio of 1:25 with either CMV-SB13 (for ChREBP) or PGK-SB13-luc (for TERT, C-myc or Jarid1B), were diluted in 2 mL of 0.9% NaCl buffer, filtered, and injected into the penis vein of seven-week-old male *mice* in 5 to 10 seconds. px330-PIK3R1, px330-axin1, px330-p53 and px330-arid1A DNA constructs, suspended in 2 mL of 0.9% NaCl saline buffer, were injected into the penis vein of seven-week-old male *mice* in 5 to 10 s. The amount of injected DNA was 60 µg for each plasmid. An equal amount of empty px330 vector was used as a control for each experiment. To follow tumor development by bioluminescence in response to the invalidation of these tumor suppressors, the PT2/C-Luc//PGK-SB13 vector was co-injected along with these plasmids at 400 ng to stably overexpress the luciferase in hepatocyte. For each experiment, the quality of hydrodynamic injection was evaluated 7 days after the injection by monitoring the presence of appropriate bioluminescent activity in liver of injected *mice*. Injected *mice* were then monitored monthly through measurement of in vivo luciferase activity using the PhotonImager system (IVIS Spectrum, PerkinElmer) and the resulting data were analyzed using the Living Image software. *Mice* were anesthetized with isoflurane and were *ip* injected with 100 mg/kg of sterile firefly D-luciferin (Caliper). After 10 min, *mice* were imaged on the PhotonImager. The number of *mice* used in each

experiment were selected based on the expected variations between animals and variability in hydrodynamic injection and SB-mediated target gene integration. No method of randomization or blinding was used in any of the experiments. Because two independent "hits" are required for tumor formation in FVB/N or C57BL/6 *mice*, only the hepatocytes that receive the two plasmids (transposon-based and transposase) will have the potential to form tumors. Therefore, *mice* that do not overexpressed the luciferase at day 7 were removed from the experimental cohort. The abdominal girth and the signs of morbidity or discomfort were monitored for all mice and tumor development was followed by echography measurement. Mice were sacrified at indicated time points or when they developed high liver tumor burden, i.e., mice showed abdominal swelling, which correlated to tumor size greater than 3 cm, and the mice were "deceased" per our Institutional Animal Care and Use Committee protocol. This maximal tumor size was not exceeded during our study.

### HCC induction with DEN/CCL4 treatment

HCC was induced in either WT or liver specific ChREBP deficient *mice* by intraperitoneal (i.p.) injection of DEN (100 mg/kg) at day 15 postpartum followed at 6 weeks old by 25 biweekly i.p injections of CCl4 (0.5 ml/kg, dissolved in corn oil). The abdominal girth and the signs of morbidity or discomfort were monitored for all mice and tumor development was followed by echography measurement. Mice were sacrificed at indicated time points or when they developed high liver tumor burden, i.e., mice showed abdominal swelling, which correlated to tumor size greater than 3 cm, and the mice were "deceased" per our Institutional Animal Care and Use Committee protocol. This maximal tumor size was not exceeded during our study.

### In vivo adenovirus injection (pre-malignant model)

$5 \times 10^8$ plaque forming units (pfu) of GFP or ChREBP adenovirus (Genecust INC) were delivered to 8 weeks old male C57BL/6J (Janvier, France) by tail vein injection as previously described[11]. To determine the contribution of p85α to ChREBP pro-proliferative effect, p85α was inhibited in ChREBP overexpressing *mice* through adenoviral delivery of specific shRNA against p85α shRNA ($5 \times 10^8$ pfu/*mice*) (Genecust INC). Effects of ChREBP overexpression with or without p85α silencing on hepatocyte proliferation were studied 3 weeks later. The number of *mice* used in each experiment were selected based on the expected variations between animals and variability in adenovirus injections. No method of randomization or blinding was used in any of the experiments. *Mice* that do not overexpressed GFP or ChREBP or do not exhibited silencing of p85α expression after adenovirus treatment were removed from the experimental cohort.

### In vivo bioluminescence imaging

After hydrodynamic injection, tumor development (increase in bioluminescence activity over time) was followed every month for 12 months. For imaging, *mice* were ip injected with 100 mg/kg of sterile firefly D-luciferin (Caliper). After 10 min, *mice* were anesthetized with isoflurane and imaged with a charge coupled device (CCD) camera (IVIS Spectrum, PerkinElmer) and the resulting data were analyzed using the Living Image software. For measurement of ChREBP transcriptional activity in vivo, the ChoRE-luc (Carbohydrate response element of the L-pyruvate kinase promoter (ChoRE) luciferase, gift from Dr M. Vasseur) plasmid was overexpressed with the RSV β-galactosidase construct through hydrodymanic gene transfer in the liver of C57BL/6J *mice* (10 µg/*mice*) as previously descried above. *Mice* were studied 48 h after injection. For imaging, *mice* were ip injected with 100 mg/kg of sterile firefly D-luciferin (Caliper). After 10 min, *mice* were imaged with the IVIS Spectrum system (PerkinElmer). For in vivo measurement of liver glucose uptake, using the BiGluc probe (SwissLumix INC), *mice* were infected with adenovirus overexpressing the luciferase ($5 \times 10^8$ pfu per *mice*). *Mice* were studied 3 weeks later

and imaged. Briefly, 24 h before imaging, *mice* were pretreated with 100 μl of 1.5 mM CLP reagent dissolved in PBS with 0.1% BSA and fasted for 12 h. The next day, *mice* were given oral gavage with 100 μl of GAz4 reagent in PBS in a dose of 12 mg/kg. The *mice* were imaged immediately using the IVIS Spectrum system (PerkinElmer) with images acquired every minute for 10 min, continuously. The resulting data were analyzed using the Living Image software.

## Mouse xenograft model

A total of $1 \times 10^7$ cells (Huh7, SNU475, SNU449 or BNL CL.2), suspended in PBS and matrigel (ratio 2:1), were injected subcutaneously into the flank of 6-week-old athymic nude *mice*. For experiments using SBI-993 (ProbeChem, #PC-61674), *mice*, inoculated with parental cells, were randomly assigned into two groups: vehicle-treated group (DMSO) and SBI-993-treated group. SBI-993 was dissolved in DMSO. *Mice* were IP injected with DMSO or SBI-993 (50 mg/kg) daily for 3 to 5 weeks depending on cell line. For each xenograft experiment, tumor width (W) and length (L) were measured each week. The tumor volume was calculated using the following formula: $L \times W2/2$. *Mice* were then sacrificed, and xenograft tumor weight was also measured.

## Sorafenib resistance xenograft model of HCC and SBI-993 treatment

Drug treatments were carried out to assess the efficacy of sorafenib alone or in combination with SBI-993 in the context of sorafenib resistance. For direct drug effect study, parental and sorafenib-resistant SNU449, SNU475, Huh7 and Sk-Hep1 were harvested and counted. $1 \times 10^7$ cells were injected subcutaneously into the flank of nude *mice*. Then, *mice* bearing indicated xenografts were orally administered 200 μl of vehicle (30% Capsitol), or 40 mg kg$^{-1}$ per day sorafenib. SBI-993 (ProbeChem, #PC-61674) was dissolved in DMSO. *Mice* were IP injected with DMSO or SBI-993 (50 mg/kg) daily. Growth of established xenografts was monitored every week by Vernier caliper measurement. For each xenograft experiment, tumor width (W) and length (L) were determined, and the tumor volume was calculated using the following formula: $L \times W2/2$.

## Hematology and blood chemistry analysis

Mouse peripheral blood was collected by cardiac puncture and placed in serum separator or dipotassium-EDTA tubes (BD Microtainer). Serum and whole blood were analyzed, the latter within 24 h after collection, by Charles River Laboratory.

## Analytical procedures

Serum ALAT, ASAT, ALP, ALB, Creatin and Bilirubin concentrations were determined using an automated Monarch device (Laboratoire de Biochimie, Faculté de Médecine Bichat, France).

## In vivo MK-2206 treatment

120 mg/kg body weight of MK-2206 (Santa Cruz Biotechnology, # 1032350-13-2) was administered orally. Captisol (Captisol) was used as a vehicle for the drug and the control animals were treated with vehicle only. MK-2206 was given orally for 3 weeks on alternate days. The dose and the duration mentioned in the study have been provided by Merck and Co.

## Tumors histopathological analysis

Liver histopathologic analysis was performed by two independent pathologists on tissue slides stained with H&E in accordance Frith criteria's[78]. In contrast to hepatocellular tumors that were usually already visible macroscopically as white nodules, preneoplastic lesions showed no expansive growth. To evaluate liver lipid content, frozen sections were subjected to Oil Red O staining using a standard protocol.

## Staining techniques

For histology studies, tissues were fixed in 10% neutral buffered formalin and embedded in paraffin. Then, 5 μm liver sections were cut, rehydrated through descending grades of ethanol and stained with hematoxylin eosin (HES) or were subjected to immunohistochemical analysis using antibodies against H&E, ChREBP (HA tag), Afp, Ki67, Akt, P-AKT, GSK3β, P-GSK3β, P70S6K and P-P70S6K according to the manufacturer's instructions. For the detection of neutral lipids, liver cryosections were stained with Oil Red O, using 0.23% of Oil Red O dye dissolved in 65% isopropyl alcohol for 10 min as previously described[11]. Hepatocyte proliferation index was also evaluated using antibody against BrdU on liver sections. BrdU (Sigma) was added to the drinking water (1 mg/mL in water) for 7 consecutive days. BrdU was replaced every 3 days.

## Mitochondrial stress test and glycolysis stress test

Mitochondrial oxygen consumption (OCR) and extracellular acidification rate (ECAR) were measured in non-buffered medium using an XF96 extracellular flux analyzer (Seahorse, Agilent, CA, USA). SNU449 and SNU475 cells were cultured in Seahorse XF RPMI assay medium at pH 7.4, supplemented with 11 mM glucose and 2 mM glutamine. 7000 SNU449 and SNU475 cells were seeded per well. After calibration of the analyzer, sequential compound injections, including oligomycin A (1 μM), carbonyl cyanide m-chlorophenyl hydrazone (CCP; 2 μM), and finally antimycin (50 μg/mL) + rotenone (10 μM) were applied to test mitochondrial respiration. Sequential compound injections, including glucose (11 mM), oligomycin A (1 μM), and 2-DG (50 mM), were applied to test glycolytic activity. OCR and ECAR values were normalized to cell DNA content.

## Metabolite analysis

GC-MS metabolomic analysis was performed by Metabolon INC from ChREBP overexpressing tumors in comparison with surrounding non-tumoral tissue (10 *mice* per group). Metabolites were extracted by 80% methanol at −20 °C and dried by vacuum centrifugation. GC-MS analysis was performed with a Waters GCT Premier mass spectrometer fitted with an Agilent 6890 gas chromatograph and a Gerstel MPS2 autosampler. Data were collected using MassLynx 4.1 software (Waters). Metabolites were identified and their peak area was recorded using QuanLynx. Data were normalized for extraction efficiency and analytical variation by mean centering the area of D4-succinate. Metabolic and lipidomic data from Metabolon INC can be found in Supplementary Data 10.

## Metabolite extraction and measurement of $^{13}$C fractional enrichments in cell samples

Cell samples were homogenized in ice-cold methanol. Metabolite extraction for LC-MS/MS analysis was prepared as previously described[79]. GC-MS was performed using an Agilent 6890N Gas Chromatograph coupled to an Agilent 5973 Mass Selective Detector (Agilent Technologies, Santa Clara, CA).

## Measurement of glucose, pyruvate and glutamine metabolic fluxes using $^{14}$C-labeled substrates

Liver explants from fed GFP or ChREBP *mice* in addition to ChREBP overexpressing tumors with corresponding non-tumoral surrounding tissues (200 mg) were incubated in duplicate in 25 mL conical glass vial sealed with rubber caps containing plastic center wells in 3 mL of Krebs-Henseleit bicarbonate buffer (pH 7.4) at 37 °C for 2 h. For glucose metabolism, oxidation, and esterification rates were assayed after incubation with 25 mM of $^{14}$C$_6$-D glucose, (Perkin Elmer, #NEC042V250UC). For glutamine metabolism, oxidation, and esterification rates were assayed after incubation with 4 mM of $^{14}$C$_5$-L glutamine (Perkin Elmer, #NEC451050UC). For pyruvate metabolism, oxidation, and esterification rates were assayed after incubation

with 0.1 mM of $^{14}C_2$-Pyruvic Acid, Sodium Salt (Perkin Elmer, #NEC256050UC).

## $^{14}CO_2$ production measurement
At the end of the incubation time, the media is transferred to a new conical glass vial for $CO_2$ production measurement and the liver explants are washed three times in ice cold PBS before processing them for lipid extraction and analysis. Perchloric acid is injected into the incubation media through the rubber cap to a final concentration of 4% (v/v). Benzethonium hydroxide is injected through the rubber cap into a plastic well suspended above the incubation media. During 1 h of vigorous shaking at 25 °C, the released $[^{14}C]$-$CO_2$ is trapped by the benzethonium hydroxide. $[^{14}C]$-$CO_2$ release is then assessed by scintillation counting.

## $^{14}C$ incorporation into intracellular lipids and TAG quantification
Fatty acid esterification rates were measured in the remaining liver piece by tracing newly synthesized triglycerides, DAG or phospholipids from $^{14}C_6$-D glucose, $^{14}C_2$-Pyruvic Acid or $^{14}C_6$-L glutamine. $^{14}C$-TGs, $^{14}C$-DAGs and $^{14}C$-PLs were extracted using chloroform/methanol (2: 1, v/v) and separated by thin-layer chromatography (TLC) on silica-gels plates (Merck Chemicals) using petroleum ether/diethyl ether/ acetic acid (85:15:0.5, v/v/v) as the mobile phase. Lipids were visualized with iodine vapor. Bands were scraped and TG, DAG and PLs were counted in scintillation liquid.

## Measurement of de novo nucleotide synthesis from glucose, glutamine and aspartate
Liver explants from fed GFP or ChREBP *mice* in addition to ChREBP overexpressing tumors with corresponding non-tumoral surrounding tissues (200 mg) were incubated in duplicate in the presence of either 25 mM of $^{14}C_6$-D glucose, (Perkin Elmer, #NEC042V250UC), 4 mM of $^{14}C_5$-L glutamine (Perkin Elmer, #NEC451050UC) or 3 mM of $^{14}C_4$-aspartic acid (Perkin Elmer, #NEC268E050UC) for 2 h. After incubation, RNAs were extracted from tissue samples using the RNeasy kit (Qiagen) based on manufacturer's instructions. RNA concentration was determined using a Nanodrop spectrophotometer. Isolated RNA was added to scintillation vials and radioactivity was measured using a liquid scintillation counter. Data was normalized to total RNA concentration.

## Lipidomic analysis
Fatty acid profiling was performed at the lipidomic core facility of Toulouse (INSERM, Metatoul). Briefly, after homogenization of tissue samples in methanol/5 mM EGTA (2:1 v/v), lipids corresponding to an equivalent of 1 mg of tissue were extracted in chloroform/methanol/ water (2.5:2.5:2.1, v/v/v), in the presence of internal standards: 1,3-dimyristine (for DAG) and glyceryl trinonadecanoate (for TG). Chloroform phases were evaporated to dryness. Neutral lipids were purified over an SPE column (Macherey Nagel glass Chromabond pure silice, 200 mg): after washing cartridge with 2 mL chloroform, lipid extract was applied on the cartridge in 20 μL chloroform, and neutral lipid were eluted with chloroform/methanol (9:1, v/v; 2 mL). The organic phase was evaporated to dryness and dissolved in 20 μL ethyl acetate. A sample (1 μL) of the lipid extract was analyzed by gas-liquid chromatography on a FOCUS Thermo Electron system, using Zebron-1 (Phenomenex) fused silica capillary columns (5 m × 0.32 mm inside diameter [i.d.], 0.50 μm film thickness). Oven temperature was programmed from 200 °C to 350 °C at a rate of 5 °C per min, and the carrier gas was hydrogen (7.25 psi). The injector and the detector were at 315 °C and 345 °C, respectively. To assess SCD1 activity, the Δ9 desaturation index was calculated as the abundance of SCD1 products (palmitoleic and oleic acids) relative to both SCD1 products and substrates (palmitic and stearic acids). Phospholipids profiling was performed by metabolon, Inc. (Durham, North Carolina, USA) from 30 mg of frozen T or NT tissues.

## Measurement of Pyruvate Dehydrogenase (PDH) activity
PDH activity was determined by measuring $^{14}C$-(1)-Pyruvic Acid oxidation from liver explants of either fed GFP/ChREBP *mice* or ChREBP overexpressing tumors with corresponding non-tumoral surrounding tissues (200 mg). Liver explants were incubated for 2 h in the presence of 0.1 mM of $^{14}C$-(1)-Pyruvic Acid, Sodium Salt (Perkin Elmer, #NEC255050UC). At the end of the incubation time, $CO_2$ release, which reflects the PDH activity, was measured as previously described above.

## Measurement of Pyruvate Carboxylase (PC) activity
Liver explants of either fed GFP/ChREBP *mice* or ChREBP overexpressing tumors with corresponding non-tumoral surrounding tissues (200 mg) were homogenized in lysis buffer containing 100 mM tris-HCl, pH 8. PC activity was then determined as described in[80]. Briefly, the oxaloacetate generated from pyruvate by the action of PC reacts with acetyl-CoA via the citrate synthase (Sigma, # C3260). The free CoA generated by this second step reacts with 5,5'-Dithiobis(2-nitrobenzoic acid) (DTNB) (Sigma, D218200). The reaction product is detected at a wavelength of 412 nm during 10 min at 30 °C. One unit (U) of pyruvate carboxylase activity is defined as the amount of enzyme required to produce 1.0 μmole of oxaloacetate in one minute.

## Measurement of cellular NADPH/NADP ratio
Total cellular NADPH/NADP ratio was determined by using the NADPH/NADP-Glo assay kit according to the manufacturer's instructions (Promega) and as previously described in[11].

## Measurement of Glucose 6-Phosphate concentrations
G6P concentrations were determined in cell extracts prepared from cultured hepatocytes or from liver samples by an enzymatic method[13].

## Measurement of HK2 enzyme Activity
$8 \times 10^6$ Huh7 cells were lysed in 500 μl of homogenization buffer consisting of 50 mM triethanolamine hydrochloride (pH 7.3), 100 mM KCl, 1 mM dithiothreitol, 5% glycerol, 1 mM EDTA, 1 mM EGTA, 1 mM phenylmethylsulfonyl fluoride, 1 μg/ml pepstatin A, 1 μg/ml leupeptin. Then HK2 activity measurement was determined as previously described in[13]. Glucose phosphorylating activity was expressed in milliunits per mg of protein.

## Culture of primary hepatocytes
Mouse hepatocytes were harvested (from WT or ChREBP KO *mice*), cultured, and infected with adenoviruses as previously described[11]. Hepatocytes were infected with adenovirus producing specific shRNA against ChREBP (ChREBPi) (1 pfu/cell) (GeneCust INC). Twenty-four hours post-infection, hepatocytes were incubated with either 5 or 25 mM glucose and 100 nM insulin for 18 h.

## Cell culture
Parental Huh7, HepG2 (ATCC, # HB-8065), SNU449 (ATCC, # CRL-2234), SNU475 (ATCC, # CRL-2236), SK-HEP-1 (ATCC, # HTB-52), BNL CL.2 (ATCC, # TIB-73) cell lines were used in this study. Huh7 cells were obtained from Dr. Perret (Cochin Institute, Paris, France)[81]. The Huh7, HepG2, SK-HEP1 and BNL CL.2 hepatoma cell lines were grown in Dulbecco's Modified Eagle's Glutamax Medium (DMEM Glutamax, glucose 4.5 g/l) containing 10% Fetal bovine serum (FBS) and 1% penicillin/streptomycin (P/S). SNU449 and SNU475 cell lines were grown in RPMI 1640 Glutamax Medium containing 10% Fetal bovine serum (FBS) and 1% penicillin/streptomycin (P/S). Cells were incubated at 37 °C in a 5% $CO_2$ atmosphere. All cell lines are tested in a regulatory basis (every 4 months) to rule out any mycoplasma contamination.

## Generation of ChREBP overexpressing or deficient cell lines
ChREBP deficient Huh7, HepG2 and BNL CL.2 cell lines were generated using a specific ChREBP shRNA producing lentivirus. This lentivirus in

addition to the control lentivirus (scramble shRNA) were produced by GeneCust (Boynes, France) using PLKO-puro lentiviral plasmid. The target sequence for *human* ChREBP was 5′ GCAGCTCCGTAAGCC CAGCA 3′. Huh7, HepG2 and BNL CL.2 cells were infected with lentivirus and selected for antibiotic resistance with 5 µg/ml of puromycin (Gibco, # A1113802). The efficiency of the ChREBP knockdown was tested by QPCR and western blotting. The *human* ChREBP SNU449 and SNU475 knock-out cell lines were generated by CRISPR/CAS9 approach using the Edit-R predesigned All-in-one lentiviral sgRNA against ChREBP from Horizon Genomics. 3 target sequences were used to generate these 2 cell lines. These target sequences for ChREBP were 5′ GCAGCTCCGTAAGCCCAGCA 3′, 5′ GGCCACCTGCCTTCCGCCTA 3′ and 5′ ACTTCATGGACATCTCAGGT 3′ localized respectively in exon 5, 6 and 7 of the gene. Both SNU449 and SNU475 ChREBP specific knockout clones were amplified and selected based on antibiotic resistance with 5 µg/ml of puromycin (Gibco, # A1113802). The efficiency of the ChREBP deletion was tested by western blotting. ChREBP overexpressing Huh7, SNU449 and SNU475 cell lines were generated by the VPR CRISPR activation (CRISPRa) approach from Horizon Genomics using specific lentiviral sgRNA targeting the promoter of ChREBP. These target sequences for the promoter of ChREBP were 5′ CAGGTGAGAACCCGGTGCTC 3′, 5′ GACTCCAAGGAAAGACGGGA 3′, 5′ CCGCCAGAGCTCCAGAGCAC 3′ and 5′ CCTTACGCCAGGTGAGA ACC 3′. P85α expression was stably deleted in ChREBP overexpressing Huh7 cell line by CRISPR/CAS9 approach using the Edit-R predesigned All-in-one lentiviral sgRNA against P85α from Horizon Genomics. 3 target sequences were used to generate this cell line. These target sequences for P85α were 5′ GTGATTATACTCTTACACTA 3′, 5′ AGAA TATACCCGCACATCCC 3′ and 5′ TTGAATTAATAAACCACTAC 3′ localized respectively in exon 3, 4 and 5 of the gene.

### Development of sorafenib resistant cells
Initially, the *human* HCC cell lines SNU449, SNU475, Huh7, and SK-Hep1 were exposed to a low dose (1 µM) of sorafenib. When the cells exhibited stable growth, we started to increase the dose of sorafenib (2, 3, 5, 7, 10 then 15 µM). The sorafenib-containing medium was replaced every two days for 8 months. The SR cells were routinely maintained under constant culture conditions including sorafenib exposure.

### Cell treatment
The following inhibitors were used in this study with indicated cell lines and culture conditions to determine how ChREBP controls cell proliferation. The PI3K/AKT signaling was inhibited with the AKT inhibitor MK-2206 at 100 nM (Santa Cruz Biotechnology, # 1032350-13-2). The G6PDH inhibitor 6-Aminonicotinamide (6AN) was used at 50 µM (Sigma, # A68203). The GLS inhibitor 6-Diazo-5-oxo-L-norleucine (DON) was used at 5 µM (Sigma, #D2141). Cells were also treated with the transaminase inhibitor amino-oxyacetic acid (AOA) at 500 µM (Sigma, # C13408). The GLUD1 inhibitor Epigallocatechin gallate (EGCG) was used at 30 µM (Sigma, # E4143). ChREBP activity was inhibited in indicated cell lines with SBI-993 at the final concentration of 40 µM (ProbeChem, # PC-61673). Sorafenib was used at the final concentration of 15 µM (Sigma, # SML2653).

### Colony formation assay
One hundred cells were seeded and cultured into 6-well plates and were allowed to grow for 2 weeks with indicated culture conditions. Colonies were then stained using 1% crystal violet solution (Sigma, V5265). The plates were then air dried and scanned at 590 nm for quantitative analysis.

### Assessment of proliferation by impedance measurements
Real-Time Cell Analysis (RTCA) was used for monitoring of cell proliferation using the iCELLigence device (Agilent Technologies, Inc). For each cell line, the optimal cell concentration was determined by serial dilution. For Huh7, HepG2, SNU-449, SNU-475 and BNL CL.2, $12 \times 10^3$ cells were seeded into each well of an E-plate L8 of the iCELLigence device. Impedance was monitored every 15 min for a period of up to 72 h. Recorded values were presented as Cell Index (CI) calculated as a relative change in the electrical impedance according to the manufacturer's instructions.

### Cell viability assay
Cells viability was assessed by the Cell Counter Kit-8 (CCK-8) assay according to the manufacturer's instructions. The absorbance at 450 nm was measured with a microplate reader. Means of 4 wells optical density (OD) in the indicated groups were used to calculated percentage of cells viability.

### Measurement of caspase 3/7 activity for apoptosis detection
Huh7, HepG2, SNU449, SNU475, SK-HEP1 and BNL CL.2 were seeded in 96-well plates for 24 h before being treated with or without sorafenib (15 µM). CASP3/7 activity was then determined by using the Caspase-Glo® 3/7 Glo Assay according to the manufacturer's protocol (Promega). CASP3/7 activity was normalized to the total number of cells per well measured at confluence. Normalized values were related to the average of untreated sample.

### Reporter assay
The ChoRE-luc (Carbohydrate response element of the L-pyruvate kinase promoter (ChoRE) luciferase, gift from Dr M. Vasseur) was used to measure ChREBP transcriptional activity in vitro. The RSV β galactosidase (RSV β-gal) plasmid was used as constitutive control reporter. Hepatocytes were transiently transfected with the appropriate combination of reporters, expression vectors, and control vectors with Lipofectamine 2000® according to the manufacturer's instructions. Twenty-four hours post-transfection, luciferase assays were performed at room temperature. Experimental data are mean of at least 3 independent experiments with luciferase activity normalized to β-galactosidase activity, conducted in triplicate.

### ChIP-sequencing
All the ChIP-Seq experiments were performed using samples collected from primary hepatocytes. ChIP-Sequencing (ChIP-seq) sample preparation and computational analysis of Illumina GA I/II data were performed by Actif Motif INC. Primary hepatocytes were incubated 24 h in the presence of 25 mM glucose and 100 nM insulin and were subjected to standard ChIP, using specific antibodies. ChIP DNA-samples were then subjected to preparation for ChIP-Seq library construction: the libraries were constructed following Illumina's Chip-Seq Sample prep kit. Briefly, Chip DNA was end-blunted and added with an 'A' base so the adaptors from Illumina with a 'T' can ligate on the ends. Then 200–400 bp fragments are gel-isolated and purified. The library was amplified by 18 cycles of PCR. Primary analysis of ChIP-Seq data sets: the image analysis and base calling were performed by using Illumina's Genome Analysis pipeline. The sequencing reads were aligned to the mouse genome UCSC build by using BOWTIE32 alignment programs in two ways: only uniquely aligned reads were kept or both uniquely aligned reads and the sequencing reads that align to repetitive regions were kept for downstream analysis (if a read aligns to multiple genome locations, only one location is arbitrarily chosen). The multiple reads were collapsed in order to reduce the PCR biases. The aligned reads were used for peak/island finding with MACS33. MACS peak/island predictions were adjusted for genome instabilities (amplifications, deletions) either by considering a local background area (MACS) that was used as a reference for the subsequent calculation of the enrichment scores. Annotating and comparing the ChIP-Seq peaks: the ChIP-Seq peaks were mapped on the UCSC genome browser. A peak was considered to be associated with a particular genome feature (for example, promoter, intron and exon) if the peak summit

(MACS peaks) was located within 3 kb distance of TSS, or within an exon or intron. If a peak intersected with multiple genome features, all the corresponding genome features were considered when computing the genome distributions. ChREBP peaks were considered common if the predicted peaks intersected over at least 1 bp. The gene ontology analysis was carried out by using DAVID/EASE35 and the sequence motif enrichment analysis was performed by using HOMER. For motif finding, we used MEME Suite with the default settings except that the expected motif site is any repetitions, and the find uncentered regions option is selected. Logos of different motifs were generated from MEME-ChIP analysis. We used Regulatory Sequence Analysis Tools to generate the final consensus sequence logo.

## Chromatin immunoprecipitation

For ChIP-seq validation, chromatin immunoprecipitation (ChIP) was performed as described[10]. Briefly, T or NT tissues from SB-mediated ChREBP overexpressing *mice* were cross-linked with 1% formaldehyde (Sigma) and chromatin DNA was sheared to 300–500 bp average in size through sonication. Resultant was immunoprecipitated with control IgG or specific antibodies overnight at 4 °C and followed by incubation with protein A/G magnetic beads (Ademtech) for an additional 2 h. After washing and elution, the protein–DNA complex was reversed by heating at 65 °C overnight. Immunoprecipitated DNA was purified by using QIAquick spin columns (Qiagen) and analyzed by qPCR using a Roche Light Cycler. Primers used are specific for regions tested and their sequences are available on request. All ChIP were repeated at least three times and representative results were shown. All signals were normalized to input chromatin signals.

## RNA profiling

The microarray experiments and data normalization were performed by the Cochin Institute transcriptomic core facility. Briefly, RNA profiling was performed using Affymetrix GeneChip *Human* Gene 2.0 ST array, which interrogates 25.000 gene sequences. Raw data were normalized using the Robust Multichip Algorithm (RMA) in Bioconductor R software. Then, all quality controls and statistics were performed using Partek GS software. First, hierarchical clustering (Pearson's dissimilarity and average linkage) and principal composant analysis (PCA) were performed as unsupervised exploratory data analysis. Then, a classical analysis of variance (ANOVA) for each gene and pair wise Tukey's post-hoc tests between groups were conducted to find differentially expressed genes. Finally, p-values and fold changes were used to filter and select differentially expressed genes. Interactions, pathways and functional enrichment analysis were carried out through the use of IPA (Ingenuity Systems, USA www.ingenuity.com) and DAVID/EASE tools (http://david.abcc.ncifcrf.gov/).

## Isolation of total RNA and analysis of mRNA expression by quantitative PCR

Total cellular RNAs from whole liver or from primary cultured hepatocytes were extracted using the RNeasy kit (Qiagen). mRNA levels were then measured as previously described using a Roche Light Cycler[10]. Primer sequences are available on request.

## Liver lysates

Livers samples (T or NT) were frozen in liquid nitrogen and kept at −80 °C until use. Mouse tissues were sonicated 3 times for 10 s each at 4 °C in lysis buffer [150 mM NaCl, 50 mM Tris-HCl pH 7.5, 5 mM EDTA, 30 mM Sodium pyrophosphate, 30 mM Sodium Fluoride, 1% Triton x 100, and protease inhibitor cocktail (Sigma, St. Louis, MO)]. Lysates were centrifuged 16.000 g at 4 °C for 20 min and supernatants were reserved for protein determination, and SDS PAGE analysis.

## Western blotting and immunoprecipitation

Western blots were carried out as previously described[10]. Revelation and quantification were performed using the Chemidoc MP system instrument (Bio-Rad). For immunoprecipitations, cell lysates (2 mg of protein) were incubated for 4 h at 4 °C with either 2 mg of anti-IGF1R or IR mAb for immunoprecipitation. Protein G agarose beads (GE Healthcare) were added and incubated at 4 °C for an additional 60 min. Beads were washed and then either resuspended or boiled in SDS sample buffer. Buffers were supplemented with 1 mM sodium orthovanadate for all steps of the assay.

## PIP3/PI(4,5)P2 quantification

Huh7 cells stably overexpressing ChREBP were seeded at density of $1.3 \times 10^7$ cells/10 cm dish. After treatment with recombinant IGF1 (10 nM) or insulin (10 nM) for 5 min, the media was removed by aspiration and 5 ml of ice-cold 0.5 M TCA was immediately added. Cells were scraped, transferred into a 15 ml tube on ice, and centrifuged at 3000 revolutions per minute (rpm) for 7 min at 4 °C. The pellet was resuspended in 3 ml of 5% tricarboxylic acid/1 mM EDTA, vortexed, and centrifuged at 3000 rpm for 5 min, the supernatant was discarded, and this washing step was repeated one more time. Afterward, neutral lipids were extracted adding 3 ml of MeOH:CHCl 3 (2:1) and continuously vortexing over 10 min at room temperature. Extracts were centrifuged at 3000 rpm for 5 min, the supernatant was discarded, and this extraction step was repeated one more time. The acidic lipids were extracted adding 2.25 ml MeOH:CHCl3:12M HCl (80:40:1) with continuous vortexing over 25 min at room temperature. Extracts were centrifuged at 3000 rpm for 5 min and the supernatant was transferred to a new 15 ml tube; 0.75 ml of CHCl 3 and 1.35 ml of 0.1 M HCl were added to the supernatant, vortexed, and centrifuged at 3000 rpm for 5 min to separate organic and aqueous phases. The organic (lower) phase was collected; 1.45 ml were transferred into new vial for PIP3 measurement and 0.05 ml were transferred into a new vial for PI(4,5)P2 measurement. All samples were dried in a vacuum dryer for 1 h. PIP3 samples were resuspended in 120 ml of PBS-Tween+3% Protein Stabilizer (provided by the Echelon kit). PI(4,5)P 2samples were resuspended in 120 ml of PBS+0.25% Protein Stabilizer. Samples were sonicated in an ice-water bath for 10 min, vortexed, and spun down before adding to the ELISA. All experiments were performed three times, each carried out in biological triplicate. Once phospholipids were isolated from cells, PIP3 and PI(4,5)P2 levels were measured using ELISA kits (Echelon, K-2500s and K4500) according to the manufacturer's instructions. The ratio between PIP3 and PI(4,5)P2 was used as an index of PI3K activity in Huh7 cells since PI(4,5)P2 levels are not influenced by PI3K activity[82].

## Statistics and reproducibility

Results are expressed as mean ± SEM and were analyzed with analysis of variance using GraphPad Prism software. Sample sizes (n) were reported in the corresponding figure legend. All experiments were performed on at least three independent occasions. No statistical method was used to predetermine sample size. After the normal distribution was confirmed with the Kolmogorov–Smirnov test, statistical comparisons between two groups were performed using a unpaired two-sided Student's *t* test followed by Mann-Whitney post hoc test. Comparisons among multiple parameters were performed by two-way ANOVA followed by Tukey's multiple-comparisons test. We did not estimate variations in the data. The variances are similar between the groups that are being statistically compared. In all cases, *P* values less than 0.05 were considered significant.

## Reporting summary

Further information on research design is available in the Nature Portfolio Reporting Summary linked to this article.

## Data availability

Source data are provided with this paper. The authors also declare that all data supporting the findings of this study are available within the paper and its supplementary information files are available from the corresponding author upon request. ChREBP, RNA PolII and H3K4me3 ChIP-seq analysis can be found at the Gene Expression Omnibus database under accession number GSE250437. Microarray analysis can be found at the Gene Expression Omnibus database under accession number GSE159517 and GSE159518. All data used for expression analysis and GSEA are publicly available. The datasets (described in Supplementary Table 1) used for determining the expression levels of ChREBP within HCC tumors in human included:

- GSE14520 (Roessler et al. [16]).
- GSE39791 (Kim et al. [17]).
- GSE57957 (Mah et al. [18]).
- GSE36376 (Lim et al. [19]).
- GSE60502 (Mas et al. [20]).
- GSE14323 (Wang et al. [21]).
- GSE6764 (Wurmbach et al. [22]).
- GSE62232 (Schulze et al. [23]).
- LIHC dataset (Oncomine database).
- LICA-FR dataset (Oncomine Database) Source data are provided with this paper.

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

## Acknowledgements

We thank the GENOM'IC, HISTIM and IMAG'IC platforms of the Cochin institute (Université Paris-Cité, Institut Cochin, INSERM, CNRS, F-75014 PARIS, France) for their help. The authors would like to particularly thank Sébastien Jacques from the Cochin institute GENOM'IC core facility for microarray and bioinformatic analysis. We grateful thank all the members of the Frédéric Bouillaud team (Institut Cochin) for helpful discussion and support of this project. The authors would like to also thank Dr Pascale Bossard for helpful discussions and critical reading of the manuscript. This work was supported by the European Research Council (ERC) under the European Union's FP7 research and innovation program (ERC-2013-StG-336629, LIPIDOLIV, R.D.), by grants from the French organizations "La Ligue contre le Cancer" (RS14/75-89, R.D.) and the "Fondation ARC" (SFI20121205630, R.D.), by the "Institut National de la Santé et de la Recherche Médicale" (Inserm), by the "Centre National de la Recherche Scientifique" (CNRS), and by the University of Paris-Cité. Funders had no role in the design of the study and data collection, analysis and interpretation, or manuscript writing. E.B. was supported by a doctoral fellowship from the University of Paris-Cité and by the French organization "La Ligue contre le Cancer". J.P. was supported by a post-doctoral fellowship from the French organization "La Ligue contre le Cancer". C.B. was supported by a post-doctoral fellowship from the European Research Council (ERC-2013-StG-336629, LIPIDOLIV, R.D.).

## Author contributions

R.D. and E.B. conceived of the hypothesis and designed the experiments. E.B., B.S., S.T., O.R., V.L., J.P., C.B., C.PO, C.PE., M.C.A.G., C.P.B., S.G. and R.D. performed the experiments. E.B. and R.D. analyzed and interpreted the data. E.B. and R.D. wrote the manuscript. All authors approved the final version of the manuscript.

## Competing interests

The authors declare no competing interests.
