## [Peer Review File · Nature Communications]

REVIEWERS' COMMENTS

Reviewer #2 (Remarks to the Author):

The manuscript by Benichou et al., The Transcription Factor ChREBP Orchestrates Liver Carcinogenesis at the Interface of the Oncogenic PI3K/AKT Signaling and Cancer Cell Metabolism, is a revised version of a manuscript that, overall, received generally positive reviews on the first submission. The authors have done a thorough and satisfactory job of addressing the comments and have completed an interesting and comprehensive study. They highlight how ChREBP can influence both signaling and metabolism to drive hepatocellular carcinoma using both human data and mice for functional and physiological studies. The work may have implications for understanding and treating HCC. No further action is required by this reviewer.

Reviewer #3 (Remarks to the Author):

The authors responded to my previous critique but not fully. It is clear that ChREBP can induce HCC and the activation of PI3K/Akt. However, the mechanism is not convincing. In contrast to the authors' interpretation overexpressed p85a was documented to inhibit PI3K activity and is considered a tumor suppressor. The authors now suggested that ChREBP is an activator of IGFR and IR. If this is the case this should be the major mechanism by which ChREBP activates PI3K. However, the mechanism by which ChREBP activate IGFR and IR remains elusive. Moreover, the activity of IGFR and IR should be measured by the phosphorylation of IRS proteins.

Reviewer #5 (Remarks to the Author):

The authors meet all the requirements requested by the reviewer.

I believe that this paper will serve as a significant reference in the field of hepatocellular carcinoma and metabolism.

RESPONSE TO REVIEWERS' COMMENTS

Reviewer #2 (Remarks to the Author):

The manuscript by Benichou et al., The Transcription Factor ChREBP Orchestrates Liver Carcinogenesis at the Interface of the Oncogenic PI3K/AKT Signaling and Cancer Cell Metabolism, is a revised version of a manuscript that, overall, received generally positive reviews on the first submission. The authors have done a thorough and satisfactory job of addressing the comments and have completed an interesting and comprehensive study. They highlight how ChREBP can influence both signaling and metabolism to drive hepatocellular carcinoma using both human data and mice for functional and physiological studies. The work may have implications for understanding and treating HCC. No further action is required by this reviewer.

Reviewer #3 (Remarks to the Author):

The authors responded to my previous critique but not fully. It is clear that ChREBP can induce HCC and the activation of PI3K/Akt. However, the mechanism is not convincing. In contrast to the authors' interpretation overexpressed p85a was documented to inhibit PI3K activity and is considered a tumor suppressor. The authors now suggested that ChREBP is an activator of IGFR and IR. If this is the case this should be the major mechanism by which ChREBP activates PI3K. However, the mechanism by which ChREBP activate IGFR and IR remains elusive. Moreover, the activity of IGFR and IR should be measured by the phosphorylation of IRS proteins.

Based on reviewer 3 comments, we provided western blots showing that activated IRS1 is increased in our pre-tumoral and tumoral model of ChREBP overexpression. This effect seems to be dependent on IRS1 since IRS2 protein content and phosphorylation are drastically decreased in response to ChREBP overexpression in our two models (data not shown). These *in vivo* data, further support our *in vitro* experiments showing in Huh7 cells that ChREBP overexpression was able to enhance PIK3 activity and subsequently PIK3/AKT signaling in response to IGF1 or insulin stimulation by stabilizing the PI3K/AKT complexes at the plasma membrane in part through controlling P85a expression. We have also changed the discussion of our manuscript to discuss in more details the fact that changes in lipid profiles at the plasma membrane (enrichment of MUFA-PL) have been demonstrated to enhance RTK signaling in different types of tumors by increasing plasma membrane fluidity. Based on our lipidomic analysis showing that ChREBP overexpressing tumor have more MUFA-PL, this could explain why IGF1R and IR activity is increased.

Reviewer #5 (Remarks to the Author):

The authors meet all the requirements requested by the reviewer.

I believe that this paper will serve as a significant reference in the field of hepatocellular carcinoma and metabolism.